# WardropNet: Traffic Flow Predictions via Equilibrium-Augmented Learning

**Kai Jungel**
School of Management
Technical University of Munich
Munich, Germany
`kai.jungel@tum.de`

**Dario Paccagnan**
Department of Computing
Imperial College London
London, United Kingdom
`d.paccagnan@imperial.ac.uk`

**Axel Parmentier**
CERMICS
École des Ponts
Marne-la-Vallée, France
`axel.parmentier@enpc.fr`

**Maximilian Schiffer**
School of Management & Munich Data Science Institute
Technical University of Munich
Munich, Germany
`schiffer@tum.de`

## Abstract

When optimizing transportation systems, anticipating traffic flows is a central element. Yet, computing such traffic equilibria remains computationally expensive. Against this background, we introduce a novel combinatorial optimization augmented neural network pipeline that allows for fast and accurate traffic flow predictions. We propose WardropNet, a neural network that combines classical layers with a subsequent equilibrium layer: the first ones inform the latter by predicting the parameterization of the equilibrium problem's latency functions. Using supervised learning we minimize the difference between the actual traffic flow and the predicted output. We show how to leverage a Bregman divergence fitting the geometry of the equilibria, which allows for end-to-end learning. WardropNet outperforms pure learning-based approaches in predicting traffic equilibria for realistic and stylized traffic scenarios. On realistic scenarios, WardropNet improves on average for time-invariant predictions by up to 72% and for time-variant predictions by up to 23% over pure learning-based approaches.

## 1 Introduction

With the advent of digitalization, big data, and sharing economies, algorithmic decision support evolved as a pivotal tool to analyze, design, and operate today's transportation systems. When designing algorithmic decision support in the context of transport optimization, an element that is crucial at all planning stages is anticipating the transportation system's response to the decision taken, e.g., to anticipate traffic flows when taking congestion pricing decisions. In general, transportation systems converge to equilibrium states in which agents cannot improve their outcome by unilaterally changing their behavior. Unfortunately, computing such equilibria usually requires a high-fidelity solver, e.g., a microscopic traffic flow simulation, that is computationally expensive and requires hours to days when computing an equilibrium state.

Against this background, we study a novel learning paradigm, that, based on knowledge of previously realized equilibria, allows us to accurately predict traffic flows in a new context within short computational times. Specifically, we assume access to contextual information $x$ that is correlated to observed traffic flows $y$, and place ourselves in a supervised learning setting: we introduce a hypothesis class $\mathcal{H}$ of *hypotheses* $h$ that map a context $x \in \mathcal{X}$ to a flow $\hat{y} \in \mathcal{Y}$, and introduce a loss $\mathcal{L}(y, \hat{y})$ that measures the difference between target flow $y$ and the predicted flow $\hat{y}$. We aim at finding the hypothesis $h \in \mathcal{H}$ that minimizes the expected loss

$$\min_{h \in \mathcal{H}} \mathbb{E}_{X,Y} \mathcal{L}\big(Y, h(X)\big). \tag{1}$$

We assume access to a training set $\{(\boldsymbol{x}_i, \boldsymbol{y}_i)\}_{i=1}^N$ derived from past traffic flow measurements or through samples obtained by a high-fidelity simulator. Based on this training set, we aim to solve the empirical supervised learning problem

$$\min_{h \in \mathcal{H}} \frac{1}{N} \sum_{i=1}^N \mathcal{L}\big(\boldsymbol{y}_i, h(\boldsymbol{x}_i)\big), \tag{2}$$

which is clearly sensitive to the chosen hypothesis class. In this context, we propose *WardropNet*, a hybrid pipeline that augments a neural network with an equilibrium layer, as such capturing combinatorial interdependencies between flows. Here, the challenge is to derive an end-to-end learning paradigm for the hyprid pipeline as this requires backpropagating the gradient through the, potentially combinatorial, equilibrium layer. Once trained, WardropNet produces efficient traffic predictions at low computational cost as inference only requires a forward pass on this neural network.

**Contribution**  In this work, we derive a combinatorial optimization-augmented machine learning (COAML) pipeline, called WardropNet, that uses a neural network to learn the parameterization of latency functions and a combinatorial optimization (CO)-layer to compute the resulting equilibrium problem. We train this pipeline via imitation learning, intuitively minimizing the Bregman divergence between target and predicted traffic flows. Crucially, we show how to train this pipeline in an end-to-end fashion by leveraging a suitably defined regularization term arising from the use of a Fenchel-Young loss. This step is key to determining a meaningful gradient, as a direct approach results in ill defined and vanishing gradients due to the combinatorial nature of the equilibrium layer. Additionally, we show that utilizing a Fenchel-Young loss in this context does not only allow to differentiate through the equilibrium layer, but it is also equivalent to minimizing the Bregman divergence between the predicted and the target traffic flows.

We present a comprehensive numerical study and show that WardropNet outperforms pure machine learning (ML) baselines on various realistic and stylized environments in both time-invariant and time-variant settings, yielding accuracy improvements of up to $75\%$. Our results show that WardropNet allows to significantly improve traffic flow predictions compared to pure ML baselines. Specifically, WardropNet benefits from the synergy between predicting latency function parameterizations with a statistical model while preserving the combinatorial structure of the respective flows within the CO-layer.

**Related work**  Our work relates to two fields: traffic equilibria prediction and CO-augmented ML. ML approaches for predicting traffic equilibria range from conventional ML approaches (Li et al., 2014; Hou et al., 2015; Lv et al., 2015) to deep learning approaches that embed physical or combinatorial solution restrictions within the network architecture (Amos & Kolter, 2017; Smith et al., 2022; Seccamonte et al., 2023). Alternatively, one can utilize graph neural networks (GNNs) which allow to embed network specific constraints (Li et al., 2018; Wu et al., 2018; Yu et al., 2018; Guo et al., 2019; Wu et al., 2019; Bai et al., 2020; Ali et al., 2022; Wu et al., 2023; Jin et al., 2024). One can also incorporate an equilibrium layer into the deep learning pipeline to directly predict equilibrium states (Bai et al., 2019; Gu et al., 2020; Li et al., 2020; Marris et al., 2022; Liu et al., 2023; McKenzie et al., 2023), which can be used to optimize stackelberg games (Sakaue & Nakamura, 2021). However, these works are either agnostic of combinatorial structures or impose major restrictions on the equilibrium layers to allow for backpropagation, e.g., the well-posedness of problems or the restriction to small quadratic programs. Contrarily, the COAML pipeline introduced in this work allows for backpropagation through general, possibly combinatorial, equilibrium layers.

Recently, COAML pipelines that embed a general CO-layer into a statistical model emerged. These pipelines aim to minimize the combinatorial error induced by the statistical model during training. First approaches train the statistical model based on the resulting combinatorial objective value (Elmachtoub et al., 2020; Elmachtoub & Grigas, 2022), which requires information about target ML predictions. Recent approaches learn the COAML pipelines in an end-to-end fashion via directly imitating combinatorial solutions, and have been applied to path problems (Parmentier, 2022) and contextual multi-stage optimization (Dalle et al., 2022; Baty et al., 2024; Jungel et al., 2024). A major challenge of these end-to-end COAML pipelines is deriving a gradient of the respective CO problem to allow for backpropagation (Agrawal et al., 2019). To do so, recent approaches introduced regularization techniques (Berthet et al., 2020) that allow to differentiate through CO problems, enabling CO problems as layers in end-to-end ML pipelines (Blondel et al., 2020). The presented

research stream shows the efficiency of COAML pipelines, but the introduced COAML pipelines have not been studied for predicting traffic equilibria. In general, equilibrium layers have neither been studied from a theoretical nor practical perspective in the context of COAML pipelines.

## 2 GENERALIZED WARDROP EQUILIBRIA

We begin by introducing the notion of wardrop equilibrium (WE), as this will be key for the development of WardropNet. In doing so, we also generalize the original notion of Wardrop (1952) to allow for the travel time on each individual arc to depend on the traffic flow on other arcs in the network. This enables us to model spill-over effects where the flow on one arc is influenced by that of neighbouring ones. Moreover, it allows us to introduce a suitable regularization term which is key for enabling end-to-end learning. For a technical derivation of the formalization, including proofs, we refer to Appendix A.

We consider a transportation network, represented as a directed graph $G = (V, A)$ with arc-set $A$ and vertex-set $V$. Here, we take the non-atomic perspective of the problem and assume that each commodity $j \in J$ has size $D^{(j)} \in \mathbb{R}_{\geq 0}$, and is required to travel from its given origin node $o^{(j)} \in V$ to its given destination node $d^{(j)} \in V$. As a result, commodity $j$ places the vector of flows $\boldsymbol{y}^{(j)} = (y_a^{(j)})_{a \in A}$ over the arcs. This flow is feasible if an amount of flow $D^{(j)}$ leaves the origin and reaches the destination while satisfying conservation of flow at all nodes. Formally, a commodity's flow $\boldsymbol{y}^{(j)} \in \mathcal{Y}^{(j)}$, lives in a (feasible) multiflow polytope $\mathcal{Y}^{(j)}$ as detailed in Appendix A. Note that we can switch between a traditional agent perspective ($D^{(j)} = 1$), which we adopt in the remainder of this paper, or a commodity perspective ($D^{(j)} \geq 1$), by restricting $D^{(j)}$ accordingly.

Finally, the traffic flow we aim to predict arises by aggregating the individual agents' flows as

$$\bar{\boldsymbol{y}} = \sum_{j \in J} \boldsymbol{y}^j \quad \text{and} \quad \bar{\mathcal{Y}} = \Big\{ \bar{\boldsymbol{y}} = \sum_{j \in J} \boldsymbol{y}^{(j)} : \boldsymbol{y}^{(j)} \in \mathcal{Y}^{(j)} \quad \forall j \in J \Big\},$$

where we note that both $\mathcal{Y}$ and $\bar{\mathcal{Y}}$ are, again, polytopes.

**Wardrop Equilibrium** Originally introduced by Wardrop (1952), a WE describes a set of flows for each agent such that any unilateral deviation would incur a longer travel time. Here the total travel time experienced by a flow is obtained by summing the travel time experience over each arc on its path. In turn, the travel time over each arc is measured by a so-called latency function, i.e., an arc-specific function $\ell_a : \mathbb{R}_+ \to \mathbb{R}_+$, mapping the aggregated flow $\bar{\boldsymbol{y}}$ in the corresponding travel time $\ell_a(\bar{\boldsymbol{y}})$. Note that this novel generalized definition allows for the travel time on each arc to depend on the aggregated flow on the entire network, and not necessarily only on the flow of that specific arc.

While there exist different, equivalent, definitions of WEs, in the following we utilize its variational characterization as derived in Appendix A.

**Definition 1** (Wardrop Equilibrium). *Given a directed graph $D = (V, A)$, origin-destination pairs $(o_j, d_j)_{j \in J}$, and a vector of latency functions $\boldsymbol{\ell} = \{\ell_a\}_{a \in A}$, a WE is a multiflow $\boldsymbol{y} = (\boldsymbol{y}^{(j)})_{j \in J} \in \mathcal{Y}^{(j)}$ such that*

$$\text{for all } j \in J \text{ and } \boldsymbol{y}'^{(j)} \in \mathcal{Y}^{(j)}, \quad \ell(\bar{\boldsymbol{y}})^\top \boldsymbol{y}^{(j)} \leq \ell(\bar{\boldsymbol{y}})^\top \boldsymbol{y}'^{(j)}. \tag{3}$$

Let us recall that our goal is to embed an equilibrium problem as a (final) layer into a neural network pipeline. In this case, it appears natural to introduce a family of latency functions and seek the best approximation among the equilibria via choosing the best latency functions. Accordingly, we aim to establish a learning algorithm that allows us to learn the parameterization of the respective latency functions. To do so, it will prove useful if our equilibrium problem remains convex.

**Convex characterization** In the decomposable case, when the latency $\tilde{\ell}_a$ on arc $a$ does only depend on the flow $\bar{y}_a$, it is well known that the WE exists and is unique, when we have non-decreasing $\tilde{\ell}_a, \forall a \in A$. We now generalize this result to the non-decomposable case. To do so, we say that latency functions $\{\ell_a\}_{a \in A}$ *derive from a potential* $\Phi : \mathbb{R}_+^a \to \mathbb{R}$ if

$$\ell_a = \frac{\partial \Phi}{\partial y_a}. \tag{4}$$

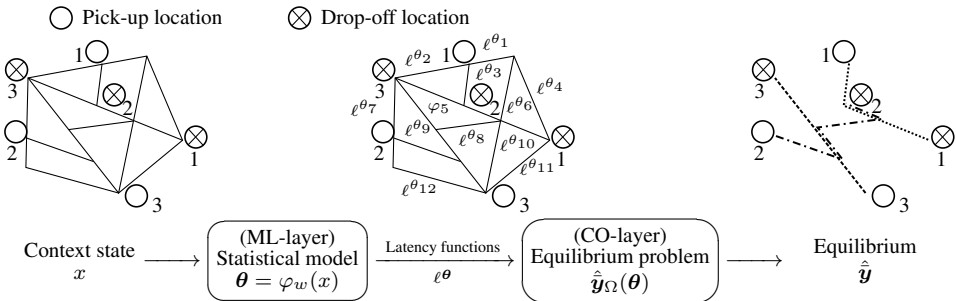

Figure 1: Schematic illustration of the WardropNet, implemented as a COAML pipeline. The pipeline comprises a statistical model $\varphi_{\boldsymbol{w}}$ and an equilibrium problem $\hat{\bar{\boldsymbol{y}}}_{\Omega}(\boldsymbol{\theta})$. The statistical model $\varphi_{\boldsymbol{w}}$ receives a context state $x$ and predicts the latency function's parameterization $\boldsymbol{\theta}$. The equilibrium problem $\hat{\bar{\boldsymbol{y}}}_{\Omega}(\boldsymbol{\theta})$ receives the latency function's parameterization $\boldsymbol{\theta}$, the transportation network, and origin-destination pairs, and yields the respective traffic equilibrium $\hat{\bar{\boldsymbol{y}}}$.

**Theorem 1** (Convex characterization). *Consider a directed graph $D = (V, A)$, origin-destination pairs $(o_j, d_j)_{j \in J}$, and a vector of latency functions $\boldsymbol{\ell} = \{\ell_a\}_{a \in A}$ that derive from a potential $\Phi$.*

1. *A multiflow $\boldsymbol{y} = (\boldsymbol{y}^{(j)})_{j \in J} \in \mathcal{Y}$ is a WE if and only if it is an optimal solution to $\min_{\boldsymbol{y} \in \mathcal{Y}} \Phi(\bar{\boldsymbol{y}}) \quad s.t. \quad \bar{\boldsymbol{y}} = \sum_j y_j$.*

2. *A vector $\bar{\boldsymbol{y}} \in \bar{\mathcal{Y}}$ is the aggregated flow of a WE if and only if it is an optimal solution to $\min_{\bar{\boldsymbol{y}} \in \bar{\mathcal{Y}}} \Phi(\bar{\boldsymbol{y}})$*

3. *If $\Phi$ is strictly convex, then a WE $\bar{\boldsymbol{y}}$ exists and is unique.*

The proof of Theorem 1 is in Appendix A. Note that Theorem 1 allows us to switch between the decomposable notion and the non-decomposable notion of latency functions.

## 3 WARDROPNET—LEARNING DATA-DRIVEN WARDROP EQUILIBRIA

Recall that our goal is to predict the aggregated traffic flow $\bar{\boldsymbol{y}}$ based on contextual information $\boldsymbol{x}$ that correlates with the observed traffic. Following the supervised learning setting from the introduction, we assume having access to samples $\{(\boldsymbol{x}_i, \bar{\boldsymbol{y}}_i)\}_{i=1}^N$, and aim at solving the resulting empirical supervised learning problem in Equation 2. In the following, we introduce our WardropNet and the end-to-end learning paradigm by first detailing the (combinatorial) hypothesis class and the respective loss function used. We then specify the resulting supervised learning problem and show that it can be interpreted as minimizing the Bregman divergence between target and predicted traffic flows.

**Hypothesis class**   Figure 1 illustrates our novel hypothesis class, which is a COAML pipeline that combines an equilibrium problem with a statistical model. To leverage this pipeline, i.e., use the statistical model $\varphi_{\boldsymbol{w}}$ to feed information that accounts for the context $\boldsymbol{x}$ into the equilibrium problem, we introduce a parameterized family of latency functions $\ell^{\boldsymbol{\theta}}$, and use the statistical model $\varphi_{\boldsymbol{w}}$ to predict its parameterization $\boldsymbol{\theta} = \varphi_{\boldsymbol{w}}(\boldsymbol{x})$.

In this context, we generally consider latency functions $\ell^{\boldsymbol{\theta}}$ that depend on $\boldsymbol{\theta}$ and derive from a *regularized potential* of the form

$$\Phi_{\boldsymbol{\theta}}(\bar{\boldsymbol{y}}) = \psi(\bar{\boldsymbol{y}}) - \boldsymbol{\theta}^{\top} \boldsymbol{\sigma}(\bar{\boldsymbol{y}})$$

where $\boldsymbol{\sigma}$ is a vector of concave basis functions, and $\psi$ is a strictly convex *regularization function* such that $\bar{\mathcal{Y}} \subseteq \mathrm{dom}(\psi)$, where $\mathrm{dom}(\psi)$ is the domain of $\psi$. Practically, we consider pipelines with linear, piecewise linear, and non-linear basis function. We choose this specific structure for $\Phi_{\boldsymbol{\theta}}$ as it mimics the structure of the Fenchel-Young loss, which will be key to obtain a differentiable end-to-end learning problem. In the following, we will first consider latency functions $\ell^{\boldsymbol{\theta}}$ that derive from a *regularized linear potential*, to ease the technical exposition. Afterward, we will provide extensions

to the general case in Section 4. Specifically, we consider

$$\Phi_{\boldsymbol{\theta}}(\bar{\boldsymbol{y}}) = \psi(\bar{\boldsymbol{y}}) - \boldsymbol{\theta}^\top \bar{\boldsymbol{y}}, \tag{5}$$

where $-\boldsymbol{\theta}$ belongs to the positive cone $\mathbb{R}_+^A$. Theorem 1 ensures that computing a WE then equals solving the following convex regularized flow problem

$$\max_{\boldsymbol{y} \in \mathcal{Y}} \boldsymbol{\theta}^\top \bar{\boldsymbol{y}} - \psi(\bar{\boldsymbol{y}}) \quad \text{where} \quad \bar{\boldsymbol{y}} = \sum_{j \in J} \boldsymbol{y}^{(j)}. \tag{6}$$

This problem can be formulated directly on the aggregated flow polytopes as

$$\hat{\bar{\boldsymbol{y}}}_\Omega(\boldsymbol{\theta}) := \arg\max_{\boldsymbol{y}} \boldsymbol{\theta}^\top \boldsymbol{y} - \Omega(\boldsymbol{y}) \quad \text{where} \quad \Omega = \psi + I_{\bar{\mathcal{Y}}} \quad \text{and} \quad I_{\bar{\mathcal{Y}}}(\boldsymbol{y}) = \begin{cases} 0 & \text{if } \boldsymbol{y} \in \bar{\mathcal{Y}}, \\ +\infty & \text{otherwise.} \end{cases} \tag{7}$$

Note that the $\arg\max$ is unique because $\Omega$ is strictly convex. This enables us to define $\hat{\bar{\boldsymbol{y}}}$ as a mapping from the latency parameterization $\boldsymbol{\theta}$ to the corresponding observed aggregated flow $\hat{\bar{\boldsymbol{y}}}(\boldsymbol{\theta})$. We can now introduce our hypothesis class formally as

$$\mathcal{H} = \left\{ h_{\boldsymbol{w}} \colon x \mapsto \hat{\bar{\boldsymbol{y}}}_\Omega \circ \varphi_{\boldsymbol{w}}(x), \, w \in \mathcal{W} \right\}.$$

To get practical hypothesis, we need to choose a regularization $\Omega$ as well as a statistical model $\varphi_{\boldsymbol{w}}$. We defer the discussion of the specific pipeline architecture to the next section, and for now focus on the loss definition.

**Fenchel-Young loss** Instead of stating a loss $\tilde{\mathcal{L}}(\bar{\boldsymbol{y}}, \hat{\bar{\boldsymbol{y}}})$ between our target traffic flow $\bar{\boldsymbol{y}}$ and the predicted traffic flow $\hat{\bar{\boldsymbol{y}}}$, we define the loss $\mathcal{L}(\boldsymbol{\theta}, \bar{\boldsymbol{y}})$ between the latency parameterization $\boldsymbol{\theta} = \varphi_{\boldsymbol{w}}(x)$ and the target traffic flow $\bar{\boldsymbol{y}}$. This loss, which is in our case the Fenchel-Young loss (Blondel et al., 2020), measures the distance between the target traffic flow $\bar{\boldsymbol{y}}$ and the traffic flow $\hat{\bar{\boldsymbol{y}}}_\Omega(\boldsymbol{\theta})$ deriving from the latency parameterization $\boldsymbol{\theta}$. The Fenchel-Young loss evolves from the Fenchel-Young inequality and leverages the relationship it describes between a convex function and its Fenchel conjugate. To understand this relationship in our context, we recall that the *Fenchel conjugate* of a function $f : \boldsymbol{y} \mapsto f(\boldsymbol{y})$ with domain $\mathcal{D}$ is defined as

$$f^*(\boldsymbol{\theta}) = \sup_{\boldsymbol{y} \in \mathcal{D}} \boldsymbol{\theta}^\top \boldsymbol{y} - f(\boldsymbol{y}).$$

Then, we can state the Fenchel conjugate of the regularization $\Omega$ as

$$\Omega^*(\boldsymbol{\theta}) = \max_{\boldsymbol{y} \in \bar{\mathcal{Y}}} \boldsymbol{\theta}^\top \boldsymbol{y} - \Omega(\boldsymbol{y}),$$

because the domain of $\Omega$ is $\bar{\mathcal{Y}}$, which is compact. Furthermore, we recall that $(\Omega^*)^* = \Omega$, because the biconjugate of a proper lower semi-continuous convex function $f$ is itself: $(f^*)^* = f$.

Then, this relationship allows us to derive a general notion of the *Fenchel-Young loss* for a regularized flow problem (6) associated to the regularization $\Omega$ as stated in Blondel et al. (2020) as

$$\mathcal{L}_\Omega(\boldsymbol{\theta}, \bar{\boldsymbol{y}}) = \Omega^*(\boldsymbol{\theta}) + \Omega(\bar{\boldsymbol{y}}) - \boldsymbol{\theta}^\top \bar{\boldsymbol{y}} = \max_{\boldsymbol{y} \in \bar{\mathcal{Y}}} \left[ \boldsymbol{\theta}^\top \boldsymbol{y} - \Omega(\boldsymbol{y}) \right] - \left[ \boldsymbol{\theta}^\top \bar{\boldsymbol{y}} - \Omega(\bar{\boldsymbol{y}}) \right]. \tag{8}$$

The first equality in (8) defines $\mathcal{L}_\Omega(\boldsymbol{\theta}, \bar{\boldsymbol{y}})$ as the left hand side of the Fenchel-Young inequality. This guarantees that $\boldsymbol{\theta} \mapsto \mathcal{L}_\Omega(\boldsymbol{\theta}, \bar{\boldsymbol{y}})$ is convex, non-negative, and equal to zero if and only if $\hat{\bar{\boldsymbol{y}}}_\Omega(\boldsymbol{\theta}) = \bar{\boldsymbol{y}}$.

**Supervised learning problem** Combining our hypothesis class with the Fenchel-Young loss, our supervised learning problem becomes

$$\min_{w} \frac{1}{N} \sum_{i=1}^{N} \mathcal{L}_\Omega\big(\varphi_{\boldsymbol{w}}(x_i), \bar{\boldsymbol{y}}_i\big), \tag{9}$$

If $\Omega^*$ is differentiable in $\boldsymbol{\theta}$, the gradient of the Fenchel-Young loss function is equal to

$$\nabla_\theta \mathcal{L}_\Omega(\boldsymbol{\theta}, \bar{\boldsymbol{y}}) = \nabla \Omega^*(\boldsymbol{\theta}) - \bar{\boldsymbol{y}}. \tag{10}$$

We later introduce regularizations $\Omega$ for which (the stochastic version of) this gradient is tractable, which enables to solve the learning problem 9 using stochastic gradient descent. For the remainder of this section we focus on showing that solving Problem 9 equals minimizing a Bregman divergence between the observed and predicted traffic flows.

**Interpretation of the Fenchel-Young loss as Bregman divergence**   Let $f : \mathcal{D} \to \mathbb{R}$ be a continuously differentiable and strictly convex function with convex domain $\mathrm{dom}(f) = \mathcal{D}$. Then, the *Bregman divergence* associated to $f$ is

$$D_f(\boldsymbol{p}, \boldsymbol{q}) = f(\boldsymbol{p}) - \big[f(\boldsymbol{q}) + \langle \nabla f(\boldsymbol{q}), \boldsymbol{p} - \boldsymbol{q} \rangle \big] \tag{11}$$

and measures the difference between two points $\boldsymbol{p}$ and $\boldsymbol{q}$ in $\mathcal{D}$ based on $f$. It is a generalization of the squared Euclidean distance $\frac{1}{2}\|\boldsymbol{p} - \boldsymbol{q}\|^2$, which we obtain when we choose $p \mapsto \frac{1}{2}\|\boldsymbol{p}\|^2$ as $f$.

Our Fenchel-Young loss $\mathcal{L}_\Omega(\boldsymbol{\theta}, \bar{\boldsymbol{y}})$ is closely related to the Bregman divergence $D_\psi$. To show this, we need the following notion. A function $\psi : \mathrm{dom}(\psi) \to \mathbb{R}$ is of *Legendre type* if its domain $\mathrm{dom}(\psi)$ is non-empty, it is strictly convex, and differentiable on the interior $\mathrm{int}(\mathrm{dom}(\psi))$ of $\mathrm{dom}(\psi)$, and $\lim\limits_{i \to \infty} \nabla \psi(\boldsymbol{p}_i) = +\infty$ for any sequence $(\boldsymbol{p}_i)$ of elements of $\mathrm{dom}(\psi)$ converging to a boundary point of $\mathrm{dom}(\psi)$. For instance, $\boldsymbol{p} \mapsto \frac{1}{2}\|\boldsymbol{p}\|^2$ is of Legendre type with domain $\mathbb{R}^d$.

Supposing that our regularization function $\psi$ is of Legendre type, which is the case for the potentials we introduce in the next section, Blondel et al. (2020, Proposition 3) show that the Fenchel-Young loss $\mathcal{L}_\Omega$ introduced above is a convex upper-bound on the Bregman divergence

$$0 \leq \underbrace{D_\psi\big(\bar{\boldsymbol{y}}, \hat{\bar{\boldsymbol{y}}}_\Omega(\boldsymbol{\theta})\big)}_{\text{possibly non-convex in } \boldsymbol{\theta}} \leq \underbrace{\mathcal{L}_\Omega(\boldsymbol{\theta}, \bar{\boldsymbol{y}})}_{\text{convex in } \boldsymbol{\theta}} \tag{12}$$

leading to equality when the loss is minimized

$$\hat{\bar{\boldsymbol{y}}}_\Omega(\boldsymbol{\theta}) = \bar{\boldsymbol{y}} \quad \Leftrightarrow \quad \mathcal{L}_\Omega(\boldsymbol{\theta}, \bar{\boldsymbol{y}}) = 0 \quad \Leftrightarrow \quad D_\psi\big(\bar{\boldsymbol{y}}, \hat{\bar{\boldsymbol{y}}}_\Omega(\boldsymbol{\theta})\big) = 0. \tag{13}$$

In the special case where $\Omega = \psi$ in addition to $\psi$ being Legendre type, we even have,

$$D_\psi\big(\bar{\boldsymbol{y}}, \hat{\bar{\boldsymbol{y}}}_\Omega(\boldsymbol{\theta})\big) = \mathcal{L}_\Omega(\boldsymbol{\theta}, \bar{\boldsymbol{y}}). \tag{14}$$

The practical consequences of Equation 12 and Equation 13 are the following: the Bregman divergence $D_\psi\big(\bar{\boldsymbol{y}}, \hat{\bar{\boldsymbol{y}}}_\Omega(\boldsymbol{\theta})\big)$ provides a natural way to measure the difference between the equilibrium we observe $\bar{\boldsymbol{y}}$ and the equilibrium we predict $\hat{\bar{\boldsymbol{y}}}_\Omega(\boldsymbol{\theta})$. While directly minimizing this Bregman divergence might lead to non-convex problems, the Fenchel-Young loss is a convex surrogate that fits this geometry and leads to a tractable learning problem.

## 4   PIPELINE ARCHITECTURE

In the following, we detail our WardropNet as COAML pipeline by first defining the statistical model $\varphi_{\boldsymbol{w}}$ before introducing various regularization techniques to obtain tractable equilibrium layers $\hat{\bar{\boldsymbol{y}}}_\Omega$.

**Statistical model**   Generally, the statistical model $\varphi_{\boldsymbol{w}}$ can be any arbitrary differentiable model which maps the context $\boldsymbol{x}$ to the latency parameterization $\boldsymbol{\theta}$. In our setting, we specify the context $\boldsymbol{x}$ as a tuple of vectors $\big((x_a)_{a \in A}, (x_v)_{v \in V}\big)$, each vector $x_a$ comprising arc specific attributes, and each vector $x_v$ comprising vertex specific attributes. The output $\boldsymbol{\theta}$ can be any set of vectors which is combinable to the latency coefficients $(\theta_{k,a})_{k \in \{1,...,K\}}^{a \in A}$, where $K$ represents the number of parameters that define the latency function on a given arc $a \in A$. The statistical model $\varphi_{\boldsymbol{w}}$ encodes its prediction via the parameterization $\boldsymbol{w}$. In Appendix B we provide a detailed description of the feedforward neural network (FNN) and GNN models used as statistical model $\varphi_{\boldsymbol{w}}$.

In the following, we focus on concisely introducing different regularization and modeling techniques to obtain tractable equilibrium layers $\hat{\bar{\boldsymbol{y}}}_\Omega$. For an in-depth discussion and a detailed derivation of the respective gradients, we refer to Appendix C.

**Latencies with Euclidean regularization**   (Details in Appendix C.1)   We can obtain a differentiable equilibrium layer by utilizing a rather simple Euclidean regularization, considering the respective squared Euclidean loss $\psi(\bar{\boldsymbol{y}}) = \frac{1}{2}\|\bar{\boldsymbol{y}}\|^2$. In this case, we retrieve decomposable affine latency functions of the form

$$\ell_a^{\boldsymbol{\theta}}(\bar{\boldsymbol{y}}) = \frac{\partial}{\partial \bar{y}_a} \Phi_{\boldsymbol{\theta}}(\bar{\boldsymbol{y}}) = -\theta_a + \bar{y}_a = \tilde{\ell}_a^{\boldsymbol{\theta}}(\bar{y}_a), \tag{15}$$

and the vector $\boldsymbol{\theta}$ corresponds to the y-intercept of the latency functions. Then, the equilibrium is the orthogonal projection of $\boldsymbol{\theta}$ on $\bar{\mathcal{Y}}$, which can be computed using any convex quadratic optimization solver.

**Regularization by perturbation** (Details in Appendix C.2) Alternatively, we can regularize our equilibrium layer by perturbation: let us consider a maximum capacity $\boldsymbol{u}$ and the corresponding aggregated flow polyhedra

$$\bar{\mathcal{Y}}_{\boldsymbol{u}} = \bar{\mathcal{Y}} \cap \{\bar{\boldsymbol{y}} \colon 0 \le \bar{\boldsymbol{y}} \le \boldsymbol{u}\}.$$

Let us now introduce a standard Gaussian vector $Z$ on $\mathbb{R}^A$. Then, we can use the convex conjugate of the expectation of the perturbed maximum flow as regularization $\Omega$,

$$\Omega = F^* \quad \text{where} \quad F(\boldsymbol{\theta}) = \mathbb{E}\Big[\max_{\boldsymbol{y} \in \bar{\mathcal{Y}}_{\boldsymbol{u}}} (\boldsymbol{\theta} + Z)^\top \boldsymbol{y}\Big]. \tag{16}$$

Berthet et al. (2020) show that this regularization yields several desirable properties: first $F$ is a proper convex continuous function, which gives $\Omega^* = (F^*)^* = F$. Second, $\text{dom}(\Omega) = \bar{\mathcal{Y}}_{\boldsymbol{u}}$, which explains why we directly defined $\Omega$ as $F^*$ and not $\psi$ in Section 3. Third, applying Danskin's lemma (Bertsekas, 2009, Proposition A.3.2), this second property, gives the differentiability of $\Omega^* = F$ and

$$\hat{\bar{\boldsymbol{y}}}_\Omega(\boldsymbol{\theta}) = \nabla\Omega^*(\boldsymbol{\theta}) = \mathbb{E}\Big[\arg\max_{\boldsymbol{y} \in \bar{\mathcal{Y}}_u} (\boldsymbol{\theta} + Z)^\top \boldsymbol{y}\Big].$$

We note that, with probability 1, the $\arg\max$ is unique on a sampled realization of $Z$. This last property is critical from a practical point of view: The exact computation of $F(\boldsymbol{\theta})$, $\hat{\bar{\boldsymbol{y}}}_\Omega(\boldsymbol{\theta})$, and $\nabla_{\boldsymbol{\theta}} \mathcal{L}_\Omega(\bar{\boldsymbol{y}})$ require to evaluate intractable integrals. However, $F(\boldsymbol{\theta})$, $\hat{\bar{\boldsymbol{y}}}_\Omega(\boldsymbol{\theta})$, and $\nabla_{\boldsymbol{\theta}} \mathcal{L}_\Omega(\bar{\boldsymbol{y}})$ all include expectations such that we can use Monte-Carlo estimations by solving the maximum flow problems for a few sampling realizations of $Z$ instead of relying on its exact computation. Lastly, we note that when using this regularization, the Bregman divergence and the Fenchel-Young loss coincide $\mathcal{L}_\Omega(\boldsymbol{\theta}, \bar{\boldsymbol{y}}) = D_\Omega(\bar{\boldsymbol{y}}, \hat{\bar{\boldsymbol{y}}}_\Omega(\boldsymbol{\theta}))$.

**Piecewise constant latencies and extended network** (Details in Appendix C.3) In practice, observed latencies often exhibit threshold effects when the traffic on an arc reaches its capacity and congestion appears, i.e., the latency increases stepwise (Vickrey, 1969). Such effects are poorly captured by our potential $\Phi_{\boldsymbol{\theta}}$ due to the smooth latency functions $\ell_a$. However, the introduced methodology can naturally be extended to piecewise constant functions of $\bar{y}_a$: let us suppose that we want to use the non-regularized latency function

$$\tilde{\ell}_a(\bar{y}_a) = \begin{cases} \tilde{\theta}_{am} & \text{if } \tau_{m-1} \le \bar{y}_a < \tau_m \text{ for } m \in \{1, \dots, M\} \\ +\infty & \text{if } \bar{y}_a > \tau_M, \end{cases}$$

where $0 = \tau_0 < \tau_1 < \dots < \tau_M$. We derive a WE considering piecewise constant latency functions by solving the non-regularized minimum cost flow $\min_{\boldsymbol{y} \in \bar{\mathcal{Y}}^{\text{ex}}} \boldsymbol{\theta}^\top \boldsymbol{y}$ on an extended network $G^{\text{exp}} = (V, A^{\text{exp}})$. The extended network $G^{\text{exp}}$ allows to encode the piecewise-constant functions in a multigraph representation with parallel arcs such that each arc $a_m$ in a set of parallel arcs with $m \in \{1, \dots M\}$ comprises a threshold cost $\tilde{\theta}_{am}$ of the original arc $a$. We can then use the regularization by perturbation approach for learning.

**Polynomial latencies regularized by perturbation** (Details in Appendix C.4) Lastly, we introduce monotonously increasing latency functions $\ell_a^{\boldsymbol{\theta}}(\bar{y}_a)$ that depend on the respective traffic flow $\bar{y}_a$. To do so, we introduce a feature mapping $\boldsymbol{\sigma}$ that maps any component $y \in \mathbb{R}_+$ to a feature vector $\boldsymbol{\sigma}(y) = (\sigma_k(y))_{k \in \{k, \dots, K\}} \in \mathbb{R}^K$ such that $y \mapsto \sigma_k(y)$ is convex for each $k$, $\boldsymbol{\sigma} \colon \boldsymbol{y} \in \mathcal{Y} \mapsto \boldsymbol{\sigma}(\boldsymbol{y}) \in \mathbb{R}^{A \times K}$. Then, we consider polynomial latency functions generated by the mapping $(\boldsymbol{\sigma}_k)_k$,

$$\ell_a^{\boldsymbol{\theta}}(\bar{y}_a) = \sum_k \theta_{k,a} \sigma_k(\bar{y}_a) \quad \text{with} \quad \theta_{k,a} \ge 0 \quad \text{for all } k \text{ and } a. \tag{17}$$

Here $\theta_{k,a}$ is the weight coefficient of the $k$-th basis function in $\ell_a$, and $\sigma_k$ is the respective basis function. The overall equilibrium is therefore parametrized by $\boldsymbol{\theta} = (\theta_{k,a})_{k \in \{1, \dots, K\}}^{a \in A}$ with $k$ indexing the mapping $(\boldsymbol{\sigma}_k)_k$. In this work, we restrict ourselves to the polynomial family $\boldsymbol{\sigma}$ using $k = \{0, 1\}$, and $\sigma_k(\bar{y}) = \bar{y}^k$. Then, the latency function for arc $a$ becomes $\ell_a^{\boldsymbol{\theta}}(\bar{y}_a) = \theta_{0,a} + \theta_{1,a} * \bar{y}_a$, with $\bar{y}_a$ denoting the aggregated flow on arc $a$. In general, we can use arbitrary complex polynomials to describe the latency function. The resulting latency function 17 is a convex function, as the potentials $(\boldsymbol{\sigma}_k)_k$ are strictly convex for each $k$. Accordingly, we can extend the regularization by perturbation to this setting as detailed in Appendix C.4.

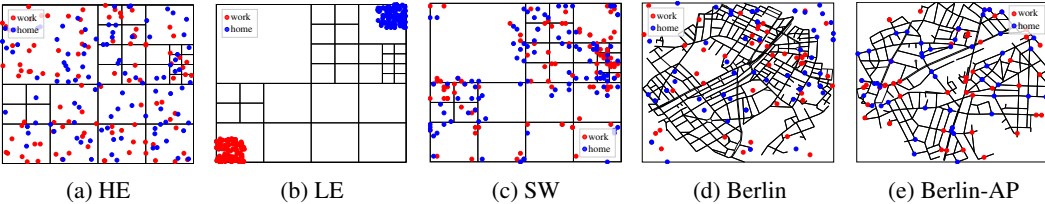

| (a) HE | (b) LE | (c) SW | (d) Berlin | (e) Berlin-AP |

Figure 2: Illustration of the considered traffic scenarios.

## 5 NUMERICAL EXPERIMENTS

In the following, we provide a concise discussion of our numerical studies. For details, we refer to Appendix D regarding the considered traffic scenarios, to Appendix $E$ regarding our COAML pipelines and baselines, and to Appendix F for an in-depth and complementary results discussions. The code for reproducing the results presented in this section is available at `https://github.com/tumBAIS/ML-CO-pipeline-TrafficPrediction`.

**Traffic Scenarios**   We consider four stylized scenarios that allow to isolate structural effects as well as two realistic scenarios to analyze the practical value of our approach. For the first two out of these six scenarios, we generate traffic samples $(x_i, y_i)_{i=1}^n$ as solutions to a static traffic equilibrium problem in the form of a WE. Instead, we produce these samples by running the widely-used dynamic traffic microsimulator *MATSim* (Horni et al., 2016) for the last four scenarios. We base our stylized scenarios, namely the *high-entropy* (HE) scenario, the *low-entropy* (LE) scenario, and the square world (SW) scenario (Figures 2a,2b,2c) on the artificial road network proposed by Eisenstat (2011) but consider different o-d pair distributions. We also consider a time-variant version (SW-TV) of the SW scenario. In the HE scenario we obtain rather distributed traffic flows and in the LE scenario rather polarized, combinatorial flows. We expect the HE scenario to be easier predictable than the LE scenario for a pure ML baseline. Moreover, we consider a realistic *Berlin* scenario (Figure 2d) that bases on a district from a realistic MATSim scenario for the city of Berlin, as well as a *Berlin-AP* scenario (Figure 2e), where we enforce all trips to artificially start and end in the respective district.

For all scenarios, we create 9 training instances, 5 validation instances, and 6 test instances. Each instance consists of a transport network with the respective target traffic flow for each road, contextual information, and origin-destination pairs. We run the training for a maximum of 20 hours, or 100 training epochs. We run the experiments on a computing cluster using 28-way Haswell-EP nodes with Infiniband FDR14 interconnect and 2 hardware threads per physical core.

**Pipelines & Baselines**   We consider two pure ML baselines, a *FNN* and a *GNN* model as detailed in Appendix B, which we trained via supervised learning directly imitating the target traffic flows $\bar{y} \in \mathbb{R}_+^{|A|}$ in the output layer. For our COAML pipelines, we leverage the very same FNN as statistical model to predict the latency parameterization $\theta$, but consider three variants for equilibrium layer as specified in Section 4: the *CL*-pipeline uses Constant Latencies regularized by perturbation, the *PL*-pipeline uses Polynomial Latencies regularized by perturbation, and the *ER*-pipeline uses latencies with an Euclidean Regularization.

Our motivation for choosing the same FNN in both the pure ML baselines and the COAML pipelines stems from the fact that we wish to compare these approaches on equal footing in a way that clearly demonstrates the impact of the additional combinatorial layer. Moreover, this choice allows us to minimize the effect that *parameter tuning* may have on the overall performance.

**Performance Analyses**   Figure 3 shows the performance of our COAML pipeline with different equilibrium layers as well as the pure ML baselines for all traffic scenarios. Focusing on our COAML pipelines, we observe that the *CL*-pipeline using piecewise constant latencies obtains the highest accuracy. The *PL*-pipeline using polynomial latencies obtains a good accuracy that is slightly worse compared to the *CL*-pipeline. The *ER*-pipeline however, falls short in performance compared to the other two variants, which might be caused by the rather simple latency functions using an

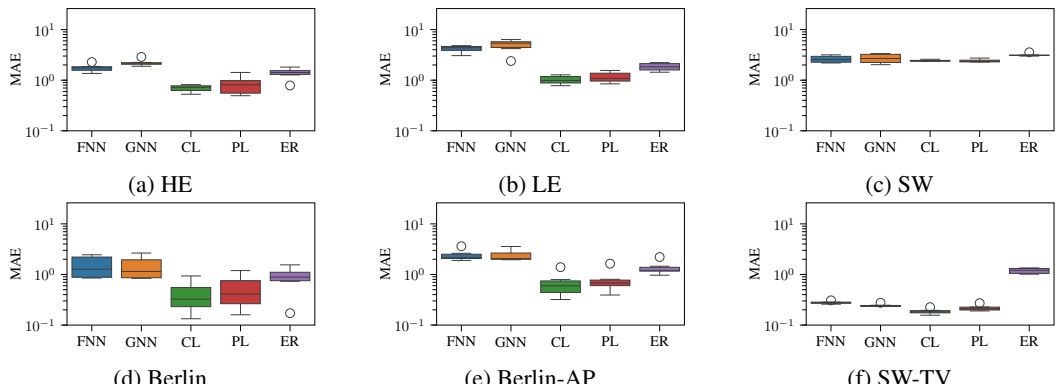

Figure 3: Benchmark performances on various stylized and realistic traffic scenarios.

**Euclidean regularization.** Focusing on the pure ML baselines, we observe that the *FNN* baseline slightly outperforms the *GNN* baseline in all time-invariant scenarios (Figures 3a-3e), while the *GNN* baseline slightly outperforms the *FNN* baseline in a time-variant scenario (Figure 3f).

For conciseness, we focus the remaining discussion of Figure 3 on comparing the best-performing COAML pipeline with the best performing ML baseline. Focusing on our stylized scenarios (Figures 3a-3c), the *CL*-pipeline outperforms the *FNN* on average by 60%, 75%, and 7% respectively. We observe that the mean absolute error (MAE) for the *FNN* increases from 1.75 in the high-entropy scenario to 4.18 in the low-entropy scenario, which indicates that the high-entropy scenario is indeed easier to predict for pure ML baselines as it has a high correlation between the traffic flow and each road's context. Remarkably, our *CL*-pipeline shows a 60% improvement over the *FNN* even in the scenario that is easier to predict for a pure ML baseline, but also shows a stable prediction error over both the high and low entropy scenario. In the square world scenario, which is again favorable for pure ML baselines due to correlation between flows and the arc features, all approaches yield a good performance. Still, the *CL*-pipeline outperforms the *FNN* baseline by 7%. Focusing on the realistic, time-invariant scenarios (Figure 3d&3e), the *CL*-pipeline outperforms the *FNN* baseline by 72% and 71% respectively. In both cases, the FNN baseline fails to learn the combinatorial structure of the respective traffic equilibrium that utilizes main roads between high demand areas stronger than small roads, while the *CL*-pipeline succeeds in encoding this structure by combining the learned latencies with the structure of the CO-layer. In the time-variant scenario (Figure 3f), the *CL*-pipeline outperforms the GNN benchmark by 23%. Note that we generally observe a lower MAE in the time-variant scenario due to an increased amount of zero-valued flows that result from the time expansion.

**Structural Analyses** Figure 4 shows an example of the structure of the predicted flows for the different algorithmic approaches for the Berlin scenario. For brevity, we exclude the visualization for the GNN and the *PL*-pipeline as they exhibit a similar structure to the *FNN* and *CL* flows, and refer to Appendix F for a complete visualization of all flows. As can be seen, the *FNN* predicts good mean values of the traffic flow but fails on predicting the true structure of the flows. In contrast, the *CL*-pipeline succeeds in predicting a realistic traffic flow with high volumes on main roads and reduced flows on smaller roads. The *ER*-pipeline fails on predicting realistic flows as the latency

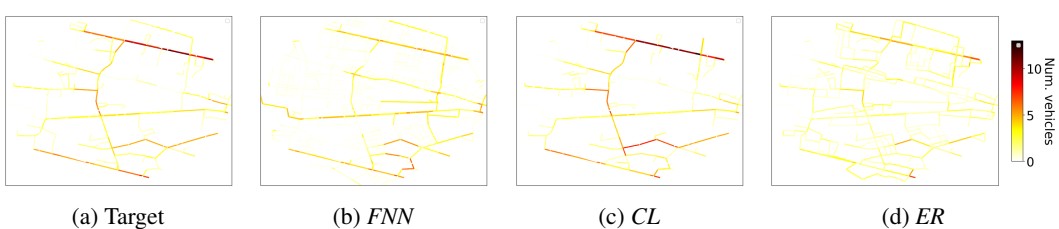

Figure 4: Visualization of time-invariant traffic flows.

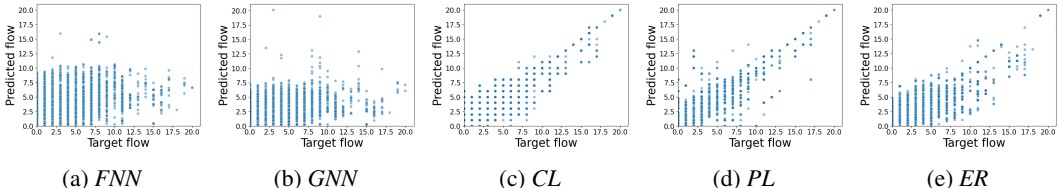

| (a) *FNN* | (b) *GNN* | (c) *CL* | (d) *PL* | (e) *ER* |

Figure 5: Comparison of target and predicted traffic flows per arc for the Berlin scenario.

function representation is too limited to inform the equilibrium layer correctly. Figure 5 supports our analyses and the findings from Figure 4 by showing the distribution of the predicted and target traffic flows, with the target traffic flows as a reference. As can be seen, the *CL* and *PL* pipelines show a higher correlation between the predicted and target traffic flows than the pure ML baselines and the *ER*-pipeline. Finally, Figure 6 shows an example of the structure of the predicted flows for the time-variant scenario. Again, we omit the *FNN* and *PL* visualizations for brevity and refer to Appendix F for a full visualization. As can be seen, the *CL*-pipeline succeeds in predicting the right traffic flow structure, while the *GNN* succeeds in predicting the right structure but underestimates the respective absolute flow volumes. The *ER*-pipeline fails in predicting both the structure and the absolute values of the respective flows.

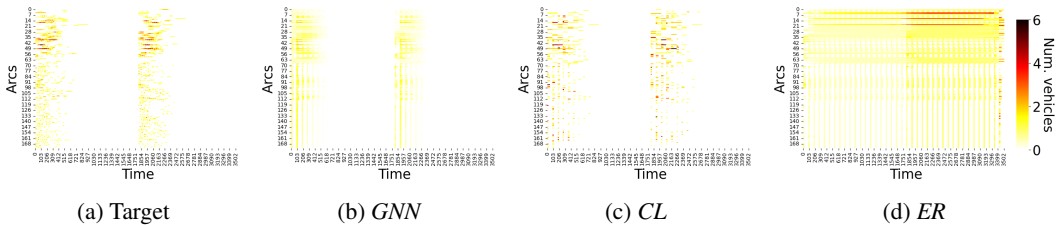

| (a) Target | (b) *GNN* | (c) *CL* | (d) *ER* |

Figure 6: Visualization of time-variant traffic flows.

## 6 CONCLUSION

In this paper we introduced WardropNet, a hybrid COAML pipeline that augments a neural network with an equilibrium layer and bases on a novel end-to-end learning paradigm. Specifically, this pipeline uses a neural network to learn the parameterization of latency functions and a CO-layer to compute the resulting equilibrium problem. We train this pipeline via imitation learning, intuitively minimizing the Bregman divergence between target and predicted traffic flows, and showed how to train this pipeline in an end-to-end fashion by leveraging a Fenchel-Young loss that allows to determine a meaningful gradient although the pipelines last layer is combinatorial. Finally, we showed with a comprehensive numerical study that our COAML pipelines outperform pure ML pipelines in various realistic scenarios by up to 72% on average. We further investigated the prediction accuracy of our COAML pipelines on time-variant traffic flows and showed that our COAML pipelines outperform pure ML baselines by up to 23% on average.

This work lays the foundation for several promising follow up works on integrating (combinatorial) equilibrium layers into neural networks for traffic flow prediction. As all of our work is open source, one can easily built upon it to account for further families of latency functions or more complex statistical models. Additionally scaling the proposed COAML pipeline to larger networks remains an interesting avenue for future research. Lastly, the presented learning paradigm is not limited to predict equilibria and can be used to learn arbitrary flow physics given the right training data and CO-layer, which opens multiple directions for further studies.

ACKNOWLEDGMENTS

This work has been funded by the Federal Ministry of Education and Research (BMBF) as part of the M-Cube Cluster 4 Future within the project DatSim, grant no. 03ZU1105LA. The work of Dario Paccagnan was supported by EPSRC grant EP/Y001001/1, funded by the International Science Partnerships Fund (ISPF) and UKRI.

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

## A GENERALIZED WARDROP EQUILIBRIA

In the following, we introduce a generalized notion of a WE, where we allow for the travel time on each individual arc to depend on the traffic flow of other arcs. This allows us to model spill-over effects where the flow on one arc is influenced by that of neighbouring ones. Moreover, it allows us to introduce a suitable regularization which is key for enabling end-to-end learning.

We consider a transportation network, represented as a directed graph $G = (V, A)$ with arc-set $A$ and vertex-set $V$. Considering a vertex $v \in V$, we denote the set of outgoing and ingoing arcs with $\delta^+(v)$, and $\delta^-(v)$. The transportation network is utilized by a set of agents $j \in J$ that travel from their origin node $o^{(j)} \in V$ to their destination node $d^{(j)} \in V$, having a demand $D^{(j)}$ of 1 in its origin and -1 in its destination node. The traveling agents induce a traffic flow equilibrium $\boldsymbol{y} = (\bar{y}_a)_{a \in A}$, formed by the emergent traffic flow $\bar{y}_a$ on all arcs $a \in A$. To formally describe this equilibrium, we denote by $\mathcal{Y} \subset \mathbb{R}^{A \times J}$ the multi-flow polytope over $D$,

$$\mathcal{Y} = \left\{ \boldsymbol{y} = (\boldsymbol{y}^{(j)})_j \colon \boldsymbol{y}^{(j)} \in \mathcal{Y}^{(j)} \right\}. \tag{18}$$

This multi-flow polytope $\mathcal{Y}$ is decomposable into flow polytope $\mathcal{Y}^{(j)} = \mathcal{F}(\boldsymbol{b}^{(j)})$ for each agent $j \in J$ with $\mathcal{Y}^{(j)} \subset \mathbb{R}^A$, and $\boldsymbol{b}^{(j)} \subset \mathbb{R}^V$. To link the respective per-agent flows to per-arc aggregated flows, we define

$$\mathcal{F}(\boldsymbol{b}) = \left\{ \boldsymbol{y} = (y_a)_{a \in A} \colon \sum_{a \in \delta^+(v)} y_a - \sum_{a \in \delta^-(v)} y_a = b_v \text{ for all } v \in V \right\}$$

$$\text{where} \quad \boldsymbol{b}^{(j)} = \left(b_v^{(j)}\right)_{v \in V} \quad \text{with} \quad b_v^{(j)} = \begin{cases} D^{(j)} & \text{if } v = o^{(j)} \\ -D^{(j)} & \text{if } v = d^{(j)} \\ 0 & \text{otherwise.} \end{cases} \quad \text{for all } v \in V. \tag{19}$$

With these definitions, we can describe solutions in the multi-flow polytope $\mathcal{Y}$ that describe the traffic flow for each agent $j \in J$. Accordingly, we can define an *aggregated flow polytope*

$$\bar{\mathcal{Y}} = \left\{ \bar{\boldsymbol{y}} = \sum_{j \in J} \boldsymbol{y}^{(j)} \colon \boldsymbol{y}^{(j)} \in \mathcal{Y}^{(j)} \text{ for all } j \text{ in } J \right\} \tag{20}$$

that describes the traffic we observe on each arc in a city's street network. Note that in this context, $\bar{\mathcal{Y}}$ is again a polytope as it results from the intersection of a sum of finitely many polytopes. In this context, it is worthwhile to remark that the inclusion

$$\bar{\mathcal{Y}} \subset \mathcal{F}\left(\sum_{j \in J} \boldsymbol{b}^{(j)}\right)$$

is, in general, a strict inclusion because the flow polytope may contain flows on paths from one agent's origin to another agent's destination. Independent of this ambiguity, it is well known that any flow in $\mathcal{Y}^{(j)}$ admits the following decomposition

$$\boldsymbol{y}^{(j)} = \sum_{P \in \mathcal{P}^{(j)}} \alpha_P^{(j)} e_P + \sum_{C \in \mathcal{C}} \beta_C^{(j)} e_C, \tag{21}$$

where $\mathcal{P}^{(j)}$ is the set of elementary $o^{(j)}$-$d^{(j)}$ paths in $G$, and $\mathcal{C}$ is the set of cycles in $G$, while $e_P$ and $e_C$ in $\{0, 1\}^A$ are the indicator vectors of $P$ and $C$, with $e_{Pa}$ (resp. $e_{Ca}$) being equal to one if $a$ belongs to $P$ (resp. $C$) and zero otherwise. Here, we observe that $\beta = 0$ in any meaningful solution as a positive $\beta$ amount to agents cycling, and w.l.o.g. ignore the $\beta$ term for the remainder of this paper. Moreover, we note that the decomposition introduced above may not be unique, i.e., one can always find a decomposition from a flow in $\mathcal{Y}^{(j)}$ to $\boldsymbol{y}^{(j)}$ but there exists no bijection that allows mapping from $\boldsymbol{y}^{(j)}$ to $\mathcal{Y}^{(j)}$. The non-existence of this bijection holds even in the absence of the $\beta$ terms.

**Wardrop Equilibrium** To find a WE in this context, we are seeking a set of flows in $\mathcal{Y}$ such that each agent travels on a path in $\mathcal{Y}^{(j)}$ where unilaterally deviating from this path would always incur a

longer travel time for the agent. In this context, the decision of one agent affects the decision of the other agents as travel times on each arc depend on its aggregated flow. To describe this dependency, we introduce arc-specific latency functions $\ell_a : \mathbb{R}_+ \to \mathbb{R}_+$ for each arc $a \in A$, which describe the travel time $\ell_a(\bar{y})$ on $a$ for an aggregated flow $\bar{y}$. Then, we can define a WE as follows.

**Definition 2** (Wardrop Equilibrium). *Given a directed graph $D = (V, A)$, origin-destination pairs $(o^{(j)}, d^{(j)})_{j \in J}$, demands $(D^{(j)})_{j \in J}$, and a vector of latency functions $\boldsymbol{\ell} = \{\ell_a\}_{a \in A}$, a multiflow $\boldsymbol{y} = (\boldsymbol{y}^{(j)})_{j \in J} \in \mathcal{Y}$ is a WE if there exists a path decomposition $\alpha^{(j)} = (\alpha_P^{(j)})_{P \in \mathcal{P}^{(j)}}, \beta^{(j)} = (\beta_C^{(j)})_{C \in \mathcal{C}^{(j)}}$ for each $j$ in $J$ such that*

$$\forall P \in \mathcal{P}^{(j)}, \ \alpha_P^{(j)} = 0 \quad if \quad P \notin \operatorname*{arg\,min}_{P \in \mathcal{P}^{(j)}} \sum_{a \in P} \ell_a(\bar{y}) \qquad and \qquad \forall C \in \mathcal{C}, \ \beta_C^{(j)} = 0 \qquad (22)$$

*where $\bar{y} = \sum_{j \in J} \boldsymbol{y}^{(j)}$ is the aggregated flow corresponding to $\boldsymbol{y}$ and $\alpha_P$ is a path decomposition.*

Our definition of a WE is a generalization of the usual notion of WE, where the latency on arc $a$ depends only on the aggregated traffic on the respective arc, i.e.,

$$\ell_a(\bar{y}) = \tilde{\ell}_a(\bar{y}_a) \quad \text{for some } \tilde{\ell}_a : \mathbb{R}_+ \to \mathbb{R}_+.$$

This generalization is meaningful for the practical application of traffic equilibrium approximators as the flow on an arc can have a spill-over effect on the flow of neighboring arcs. Thus, considering the flow of neighboring arcs for the approximation of the traffic flow on a specific arc might ease accurate predictions. From a theoretical perspective, we will later show that this generalized formulation of WEs allows us to equip latency functions with regularization terms which is crucial to derive an end-to-end learning framework.

**Variational characterization:** To establish our learning-based approximation, we are interested in obtaining a more aggregated notion of a WE. To this end, we show that one can define the condition for a Wardrop equilibrium without using the path decomposition of $y_j$.

**Proposition 1** (Wardrop Equilibrium (Variational)). *Given a directed graph $D = (V, A)$, origin-destination pairs $(o_j, d_j)_{j \in J}$, and a vector of latency functions $\boldsymbol{\ell} = \{\ell_a\}_{a \in A}$, a multiflow $\boldsymbol{y} = (\boldsymbol{y}^{(j)})_{j \in J} \in \mathcal{Y}$ is a WE if and only if*

$$\text{for all } j \in J \text{ and } \boldsymbol{y}'^{(j)} \in \mathcal{Y}^{(j)}, \quad \boldsymbol{\ell}(\bar{\boldsymbol{y}})^\top \boldsymbol{y}^{(j)} \le \boldsymbol{\ell}(\bar{\boldsymbol{y}})^\top \boldsymbol{y}'^{(j)} \qquad (23)$$

*where $\mathcal{Y}^{(j)}$ represents the flow polytope of Agent $j$.*

This flow polytope $\mathcal{Y}^{(j)} \subseteq \mathbb{R}^{|A|}$ ensures that any feasible solution $y_j$ is non-negative and sends one unit of flow from origin $o_j$ to destination $d_j$, while ensuring mass conservation at the nodes of the network $G(V, A)$.

*Proof.*
i) Forward direction: We first prove that 22 implies 23.

Let $A = \min_{P \in \mathcal{P}^{(j)}} \sum_{a \in P} \ell_a(\bar{y})$. Then, 21 gives the decompositions $y'^{(j)} = \sum_{P \in \mathcal{P}^j} \alpha'_P e_P$ and $y^{(j)} = \sum_{P \in \mathcal{P}^j} \alpha_P e_P$. Taking the dot product with $\ell(\bar{y})$ gives

$$\ell(\bar{y})^\top (y'^{(j)}) = \ell(\bar{y})^\top \Big( \sum_{P \in \mathcal{P}^{(j)}} \alpha'_P e_P \Big) = \sum_{P \in \mathcal{P}^{(j)}} \alpha'_P e_P^\top \ell(\bar{y}) \ge D^{(j)} A$$

$$\ell(\bar{y})^\top (y^{(j)}) = \ell(\bar{y})^\top \Big( \sum_{P \in \mathcal{P}^{(j)}} \alpha_P e_P \Big) = \sum_{P \in \mathcal{P}^{(j)}} \alpha_P \underbrace{e_P^\top \ell(\bar{y})}_{=A} = D^{(j)} A$$

which gives $\ell(\bar{y})^\top (y^{(j)}) \le \ell(\bar{y})^\top (y'^{(j)})$.

ii) Backward direction: Let us now prove that 23 implies 22.

To that purpose, consider a $\boldsymbol{y}$ such that Equation 22 is not satisfied. We prove the result by exhibiting $\boldsymbol{y}'^{(j)}$ such that $\boldsymbol{\ell}(\bar{\boldsymbol{y}})^\top \boldsymbol{y}^{(j)} > \boldsymbol{\ell}(\bar{\boldsymbol{y}})^\top \boldsymbol{y}'^{(j)}$. Let $\mathcal{P}^{\text{o}} = \operatorname*{arg\,min}_{P \in \mathcal{P}^{(j)}} \boldsymbol{\ell}(\bar{\boldsymbol{y}})^\top e_P$ and $A = \min_{P \in \mathcal{P}^{(j)}} \boldsymbol{\ell}(\bar{\boldsymbol{y}})^\top e_P$. We have

$$\ell(\bar{y})^\top (y^{(j)}) = \sum_{P \in \mathcal{P}^{\text{o}}} \alpha_P \underbrace{\boldsymbol{\ell}(\bar{\boldsymbol{y}})^\top e_P}_{=A} + \sum_{P \in \mathcal{P}^{(j)} \setminus \mathcal{P}^{\text{o}}} \alpha_P \underbrace{\boldsymbol{\ell}(\bar{\boldsymbol{y}})^\top e_P}_{>A} > D^{(j)} A,$$

with the last inequality being true because $\sum_{P \in \mathcal{P}^{(j)} \setminus \mathcal{P}^\circ} \alpha_P > 0$ by hypothesis and $\sum_{P \in \mathcal{P}^{(j)}} \alpha_P = D^{(j)}$ because $\boldsymbol{y}^{(j)} \in \mathcal{Y}^{(j)}$. Let us now pick a $P$ in $\mathcal{P}^\circ$, and set $\boldsymbol{y}'^{(j)} = D^{(j)} \boldsymbol{e}_P$. We have $\boldsymbol{y}'^{(j)} \in \mathcal{P}^{(j)}$ and $\boldsymbol{\ell}(\bar{\boldsymbol{y}})^\top \boldsymbol{y}^{(j)} > D^{(j)} A = \boldsymbol{\ell}(\bar{\boldsymbol{y}})^\top \boldsymbol{y}'^{(j)}$, which concludes the proof. $\qquad\square$

Here, the latency function $\ell_a : \mathbb{R}_{\geq 0} \to \mathbb{R}_{\geq 0}$ describes the travel time $t_a = \ell_a(\bar{y}_a)$ on arc $a$ as a function of its aggregated flow $\bar{y}_a = \sum_{j \in J} y_a^{(j)}$. Clearly, a WE in line with Proposition 1 is sensitive to the respective latency functions in $\boldsymbol{\ell}$.

Let us recall that our goal is to approximate an equilibrium by a surrogate, which, in our case, is a learning-enriched WE. In this case, it appears natural to introduce a family of latency functions and seek the best approximation among the equilibria generated by this family. Accordingly, we aim to establish a learning algorithm that allows us to learn the parameterization of the respective latency functions. To do so, it will prove useful if our equilibrium problem remains convex.

**Convex characterization**  In the decomposable case, it is well known that the WE exists, is unique, and is characterized when all $\tilde{\ell}_a$ are non-decreasing. We now generalize this result to the non-decomposable case. To do so, we consider latency functions $(\ell_a)_a$ that *derive from a potential* $\Phi : \mathbb{R}_+^a \to \mathbb{R}$ if

$$\ell_a = \frac{\partial \Phi}{\partial y_a}. \tag{24}$$

**Theorem 1** (Convex characterization). *Consider a directed graph $D = (V, A)$, origin-destination pairs $(o_j, d_j)_{j \in J}$, and a vector of latency functions $\boldsymbol{\ell} = \{\ell_a\}_{a \in A}$ that derive from a potential $\Phi$.*

1. *A multiflow $\boldsymbol{y} = (\boldsymbol{y}^{(j)})_{j \in J} \in \mathcal{Y}$ is a WE if and only if it is an optimal solution to*

$$\min_{\boldsymbol{y} \in \mathcal{Y}} \Phi(\bar{\boldsymbol{y}}) \quad s.t. \quad \bar{\boldsymbol{y}} = \sum_j y_j. \tag{25}$$

2. *A vector $\bar{\boldsymbol{y}} \in \bar{\mathcal{Y}}$ is the aggregated flow of a WE if and only if it is an optimal solution to*

$$\min_{\bar{\boldsymbol{y}} \in \bar{\mathcal{Y}}} \Phi(\bar{\boldsymbol{y}}) \tag{26}$$

3. *If $\Phi$ is strictly convex, then a WE $\bar{\boldsymbol{y}}$ exists and is unique.*

*Proof.* First, we prove that a flow is a user equilibrium if and only if it satisfies the first-order conditions of the optimization problem, even if it is not convex. Using the path variables $\alpha_P$ of Equation 21, we can rewrite the convex optimization problem,

$$\min_{\boldsymbol{y} \in \mathcal{Y}} \Phi(\bar{\boldsymbol{y}}) \quad \text{s.t.} \quad \bar{\boldsymbol{y}} = \sum_j y_j. \tag{27}$$

as follows:

$$\min_{\alpha} \Phi\Big( \sum_{P \in \mathcal{P}} \alpha_P \boldsymbol{e}_P \Big) \tag{28a}$$

$$s.t. \, D^{(j)} = \sum_{P \in \mathcal{P}^{(j)}} \alpha_P \qquad \forall j \in J \tag{28b}$$

$$\alpha_P \geq 0 \qquad \forall P \in \mathcal{P} \tag{28c}$$

with $\mathcal{P}$ being the union of the disjoint $\mathcal{P}^{(j)}$. Note that the agent-specific paths $\mathcal{P}^{(j)}$ remain disjoint, as we model each agent with a unique origin-destination pair.

Denoting by $\lambda_j$ and $\mu_P$ the dual variables of constraints 28b and 28c, we obtain the Lagrangian

$$\mathcal{L}(\boldsymbol{y}, \boldsymbol{\lambda}, \boldsymbol{\mu}) = \Phi\Big( \sum_{P \in \mathcal{P}^{(j)}} \alpha_P \boldsymbol{e}_P \Big) + \sum_{j=1}^J \lambda_j \Big( D^{(j)} - \sum_{P \in \mathcal{P}^{(j)}} \alpha_P \Big) + \sum_{P \in \mathcal{P}} \mu_P \alpha_P$$

We have

$$\frac{\partial \Phi(\bar{\boldsymbol{y}})}{\partial \alpha_P} = \sum_{a \in P} \frac{\partial \Phi(\bar{\boldsymbol{y}})}{\partial y_a} = \sum_{a \in P} \ell_a(\bar{\boldsymbol{y}}) = \ell_P(\bar{\boldsymbol{y}}) \quad \text{which gives} \quad \frac{\partial \mathcal{L}(\boldsymbol{y}, \boldsymbol{\lambda}, \boldsymbol{\mu})}{\partial \alpha_P} = \ell_P(\bar{\boldsymbol{y}}) - \lambda_j + \mu_P.$$

The KKT conditions for 28 are therefore

$$\frac{\partial \mathcal{L}(\boldsymbol{y}, \boldsymbol{\lambda}, \boldsymbol{\mu})}{\partial \alpha_P} = \ell_P(\bar{\boldsymbol{y}}) - \lambda_j + \mu_P = 0 \qquad \forall P \in \mathcal{P}^{(j)}, \forall j \in J$$

$$\frac{\partial \mathcal{L}(\boldsymbol{y}, \boldsymbol{\lambda}, \boldsymbol{\mu})}{\partial \mu_j} = D^{(j)} - \sum_{P \in \mathcal{P}^{(j)}} \alpha_P = 0 \qquad \forall j \in J$$

$$\mu_P = 0 \text{ or } \alpha_P = 0 \qquad \forall P \in \mathcal{P}$$

$$\mu_P \leq 0, \alpha_P \geq 0 \qquad \forall p \in \mathcal{P}$$

Then a flow $\bar{\boldsymbol{y}}$ satisfies the KKT conditions if it satisfies for all origin-destination pairs $j \in J$ and all paths $P \in \mathcal{P}^{(j)}$

$$\ell_P(\bar{\boldsymbol{y}}) = \lambda_j \quad \text{if } \alpha_P > 0$$
$$\ell_P(\bar{\boldsymbol{y}}) \geq \lambda_j \quad \text{if } \alpha_P = 0$$

If the path $P \in \mathcal{P}^{(j)}$ is used, then its cost is $\lambda_j$, and all other paths $P' \in \mathcal{P}^{(j)}$ have a greater or equal cost. This is just the characterization of a WE in Definition 2.

The characterization of aggregate flows in the second point of the theorem is a direct consequence of the first and of the definition of $\bar{\mathcal{Y}}$.

Existence and uniqueness of the solution $\bar{\boldsymbol{y}}$ come from the existence (and uniqueness) of the optimum of a (strictly) convex optimization problem on a convex polytope (which is also compact). $\square$

Theorem 1 allows us to switch between the decomposable notion of latency functions and a non-decomposable notion of latency functions as shown in Example 1.

**Example 1.** *If $\ell$ is decomposable, $\ell_a(\bar{\boldsymbol{y}}) = \tilde{\ell}_a(\bar{y}_a)$, then defining*

$$L_a(\bar{y}_a) = \int_0^{y_a} \tilde{\ell}_a(u) \, du$$

*we get that $\ell$ derives from the potential*

$$\Phi(\bar{\boldsymbol{y}}) = \sum_a L_a(\bar{y}_a).$$

*And Theorem 1 becomes the classic characterization of WE as a convex optimization problem.*

## B STATISTICAL MODELS

**FNN:** Our FNN model (Figure 7) comprises a set of parallel multi layer perceptrons (MLPs). Each MLP receives as input a vector of features $x_a$ describing the attributes of arc $a$ and outputs an embedding vector. Each MLP consists of 5 layers, with [100, 500, 100, 10, 5] nodes. We consider *relu* activation functions between the layers. If the MLP is the last module in the pipeline there is an output layer of 1 node convoluting the embedding layer to the output dimension. The activation function in the output layer is pipeline dependent: In the case, when the FNN directly outputs $\boldsymbol{y}$ the last dense layer does not consider any activation function. In the case, when the FNN outputs the latency parameters $(\varphi_{\boldsymbol{w}}^{\text{FNN}}(x_a) = (\theta_{k,a})_{k \in \{1,\ldots,K\}})_{a \in A} = (\theta_{k,a})_{k \in \{1,\ldots,K\}}^{a \in A}$, then the last dense layer considers a *softplus* activation function. We specify the considered features in Appendix B.1.

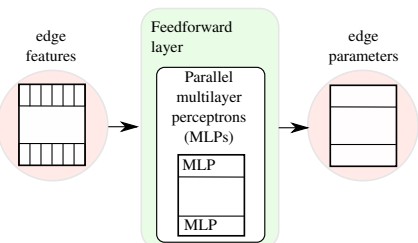

Figure 7: FNN model.

**GNN:** Our GNN model considers a graph message passing module and two subsequent FNN modules (Figure 8). The message passing module receives as input the set of arc feature vectors $(x_a)_{a \in A}$ and the set of vertex feature vectors $(x_v)_{v \in V}$. The GNN applies 3 message passing convolutions and outputs the latent node embeddings. Here, each convolution considers embeddings of length 20 and feedforward layers with [20, 20] nodes with *relu* activation functions. Then, the GNN inputs the combined feature sets of latent vertex embeddings from arc-starting nodes, and original arc features into an FNN that outputs the arc embeddings. Subsequently, the GNN inputs the arc embeddings into an FNN to output the arc parameters. We specify the considered features in Appendix B.1.

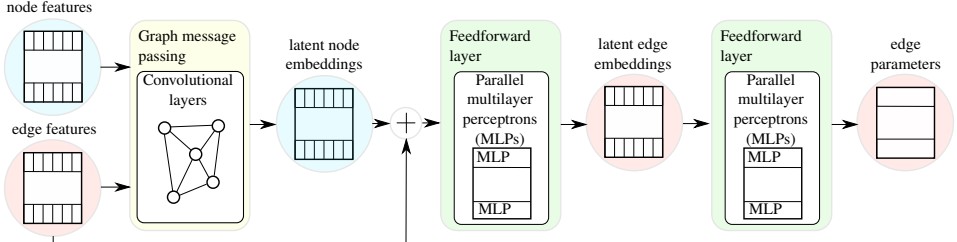

Figure 8: GNN model.

### B.1  FEATURES

In Table 1 we summarize the features considered in the COAML pipeline benchmarks and the *FNN* benchmark. In Table 2 we summarize the features considered in the *GNN* benchmark. The area factor describes the radius considered for computing the features: $\frac{\text{Max distance between nodes in network}}{\text{Area Factor}}$. We standardize all features with its mean values.

Table 1: Features considered in the *FNN*.

| Arc features |
| --- |
| number of home locations (area factors: 1; 2; 5; 10; 15) |
| number of work locations (area factors: 1; 2; 5; 10; 15) |
| number of nodes (area factors: 1; 2; 5; 10; 15) |
| number of arcs (area factors: 1; 2; 5; 10; 15) |
| number of arcs with capacity ¿ 1000 (area factors: 1; 2; 5; 10; 15) |
| arc length |
| arc speed |
| arc capacity |
| arc number of lanes |
| arc number of lanes |
| arc transit time |
| number of starting arcs at starting node |
| number of ending arcs at starting node |
| number of starting arcs at ending node |
| number of ending arcs at ending node |

| Arc features when learning time-variant flow |
| --- |
| distance from time morning rush hour in seconds |
| (distance from time morning rush hour in seconds)$^2$ |
| (distance from time morning rush hour in seconds)$^3$ |
| remaining time in seconds |
| simulation time in seconds |
| distance time from start evening rush hour in seconds |
| (distance time from start evening rush hour in seconds)$^2$ |
| (distance time from start evening rush hour in seconds)$^3$ |

| Arc features when learning piecewise constant latencies |
| --- |
| capacity index $\zeta$ of arc |

Table 2: Features considered in the *GNN*.

| Arc features |
| --- |
| arc length |
| arc speed |
| arc capacity |
| arc number of lanes |
| arc number of lanes |
| arc transit time |
| number of starting arcs at starting node |
| number of ending arcs at starting node |
| number of starting arcs at ending node |
| number of ending arcs at ending node |

| Arc features when learning time-variant flow |
| --- |
| distance from time morning rush hour in seconds |
| (distance from time morning rush hour in seconds)$^2$ |
| (distance from time morning rush hour in seconds)$^3$ |
| remaining time in seconds |
| simulation time in seconds |
| distance time from start evening rush hour in seconds |
| (distance time from start evening rush hour in seconds)$^2$ |
| (distance time from start evening rush hour in seconds)$^3$ |

| Node features |
| --- |
| number of home locations (area factors: 1; 2; 5; 10; 15) |
| number of work locations (area factors: 1; 2; 5; 10; 15) |
| number of nodes (area factors: 1; 2; 5; 10; 15) |
| number of arcs (area factors: 1; 2; 5; 10; 15) |
| number of arcs with capacity ¿ 1000 (area factors: 1; 2; 5; 10; 15) |

| Node features when learning time-variant flow |
| --- |
| simulation time in seconds |

## C EQUILIBRIUM LAYERS

In this section, we provide background on the equilibrium layers introduced in Section 4.

### C.1 BACKGROUND ON LATENCIES WITH EUCLIDEAN REGULARIZATION

We can obtain a differentiable equilibrium layer by utilizing a rather simple Euclidean regularization, considering the respective squared Euclidean loss

$$\psi(\bar{\boldsymbol{y}}) = \frac{1}{2}\|\bar{\boldsymbol{y}}\|^2.$$

In this case, we retrieve decomposable affine latency functions of the form

$$\ell_a^{\boldsymbol{\theta}}(\bar{\boldsymbol{y}}) = \frac{\partial}{\partial \bar{y}_a}\Phi_{\boldsymbol{\theta}}(\bar{\boldsymbol{y}}) = -\theta_a + \bar{y}_a = \tilde{\ell}_a^{\boldsymbol{\theta}}(\bar{y}_a), \tag{29}$$

and the vector $\boldsymbol{\theta}$ corresponds to the y-intercept of the latency functions. In this case, the equilibrium problem 6 becomes

$$\hat{\bar{\boldsymbol{y}}}_\Omega(\boldsymbol{\theta}) = \arg\max_{\boldsymbol{y}\in\bar{\mathcal{Y}}} \boldsymbol{\theta}^\top \boldsymbol{y} - \frac{1}{2}\|\boldsymbol{y}\|^2 = \arg\max_{\boldsymbol{y}\in\bar{\mathcal{Y}}} \frac{1}{2}\|\boldsymbol{\theta} - \boldsymbol{y}\|^2. \tag{30}$$

In other words, the equilibrium is the orthogonal projection of $\boldsymbol{\theta}$ on $\bar{\mathcal{Y}}$ and can be efficiently computed using any convex quadratic optimization solver. In the learning problem 9, we need the gradient $\nabla\Omega^*(\boldsymbol{\theta})$ of $\Omega^*(\boldsymbol{\theta}) = \max_{\boldsymbol{y}\in\mathcal{Y}} \boldsymbol{\theta}^\top \boldsymbol{y} - \frac{1}{2}\|\boldsymbol{y}\|^2$. Danskin's lemma ensures that

$$\nabla\Omega^*(\boldsymbol{\theta}) = \arg\max_{\boldsymbol{y}\in\bar{\mathcal{Y}}} \boldsymbol{\theta}^\top \boldsymbol{y} - \frac{1}{2}\|\boldsymbol{y}\|^2 = \hat{\bar{\boldsymbol{y}}}_\Omega(\boldsymbol{\theta}).$$

### C.2 BACKGROUND ON REGULARIZATION BY PERTURBATION

Alternatively, we can regularize our equilibrium layer by perturbation: let us a consider a maximum capacity $\boldsymbol{u}$ and the corresponding aggregated flow polyhedra

$$\bar{\mathcal{Y}}_{\boldsymbol{u}} = \bar{\mathcal{Y}} \cap \big\{\bar{\boldsymbol{y}} : 0 \leq \bar{\boldsymbol{y}} \leq \boldsymbol{u}\big\}.$$

Let us now introduce a standard Gaussian vector $Z$ on $\mathbb{R}^A$. Then, we can use the convex conjugate of the expectation of the perturbed maximum flow as regularization $\Omega$,

$$\Omega = F^* \quad \text{where} \quad F(\boldsymbol{\theta}) = \mathbb{E}\Big[ \max_{\boldsymbol{y} \in \bar{\mathcal{Y}}_u} (\boldsymbol{\theta} + Z)^\top \boldsymbol{y} \Big]. \tag{31}$$

Berthet et al. (2020) show that this regularization yields several desirable properties: first $F$ is a proper convex continuous function, which gives $\Omega^* = (F^*)^* = F$, second, $\text{dom}(\Omega) = \bar{\mathcal{Y}}_{\boldsymbol{u}}$. Third, applying Danskin's lemma, this second property, gives the differentiability of $\Omega^* = F$ and

$$\hat{\bar{\boldsymbol{y}}}_\Omega(\boldsymbol{\theta}) = \nabla \Omega^*(\boldsymbol{\theta}) = \mathbb{E}\Big[ \arg\max_{\boldsymbol{y} \in \bar{\mathcal{Y}}_u} (\boldsymbol{\theta} + Z)^\top \boldsymbol{y} \Big].$$

We note that the $\arg\max$ is unique with probability $1$ on the sampling of $Z$. This last property is critical from a practical point of view. Indeed, while the exact computation of $F(\boldsymbol{\theta})$, $\hat{\bar{\boldsymbol{y}}}_\Omega(\boldsymbol{\theta})$, and $\nabla_{\boldsymbol{\theta}} \mathcal{L}_\Omega(\bar{\boldsymbol{y}})$ require to evaluate intractable integrals, they are all expectations for which Monte-Carlo estimations can be computed by sampling a few realizations of $Z$ and computing the corresponding maximum flow problem.

Finally, Bregman divergence and Fenchel-Young loss coincide for this regularization $\mathcal{L}_\Omega(\boldsymbol{\theta}, \bar{\boldsymbol{y}}) = D_\Omega(\bar{\boldsymbol{y}}, \hat{\bar{\boldsymbol{y}}}_\Omega(\boldsymbol{\theta}))$. This follows from Equation 14, even though $\Omega$ is not Legendre type as a function on $\mathbb{R}^A$, because it is not even differentiable since its domain $\bar{\mathcal{Y}}_u$ is not full dimensional. Indeed, if we consider the restriction of $F$ and $\Omega$ to the direction of the affine hull of $\bar{\mathcal{Y}}_{\boldsymbol{u}}$, which is equal to the linear span of $\bar{\mathcal{Y}}_{\boldsymbol{u}}$ since $0 \in \bar{\mathcal{Y}}_{\boldsymbol{u}}$, then $\Omega$ is Legendre type. Equation 14 gives the equality on the linear span, which can be extended to $\mathbb{R}^A$ since the component of $\boldsymbol{\theta}$ in the orthogonal of the linear span does not impact a role in the $\arg\max$.

### C.3 BACKGROUND ON PIECEWISE CONSTANT LATENCIES AND EXTENDED NETWORK

In practice, observed latencies often exhibit threshold effects when the traffic on an arc reaches its capacity and congestion appears, i.e., the latency increases stepwise (Vickrey, 1969). Such effects are poorly captured by our potential $\Phi_{\boldsymbol{\theta}}$ due to the smooth latency functions $\ell_a$.

However, the methodology that we introduced can naturally be extended to the case where such a latency function can be modeled by piecewise constant functions of $\bar{y}_a$: let us suppose that we want to use the non-regularized latency function

$$\tilde{\ell}_a(\bar{y}_a) = \begin{cases} \tilde{\theta}_{a,m} & \text{if } \tau_{m-1} \le \bar{y}_a < \tau_m \text{ for } m \in \{1, \dots, M\} \\ +\infty & \text{if } \bar{y}_a > \tau_M, \end{cases}$$

where $0 = \tau_0 < \tau_1 < \dots < \tau_M$. Such latency functions are in line with traffic physics: an arc might allow free floating traffic till a certain threshold capacity is reached. Then the latency increases stepwise, e.g., with the traffic experiencing reduced speed, stop-and-go, and finally, congestion (Vickrey, 1969).

We derive piecewise constant latency functions by solving the non-regularized WE as a minimum cost flow

$$\min_{\boldsymbol{y} \in \mathcal{Y}^{\text{ex}}} \boldsymbol{\theta}^\top \boldsymbol{y}$$

on an extended network $G^{\text{exp}} = (V, A^{\text{exp}})$ that allows to encode the piecewise-constant functions in a multi-graph representation with parallel arcs.

Specifically, we construct the extended network $G^{\text{exp}} = (V, A^{\text{exp}})$ based on the transporation network $G = (V, A)$ as follows: Vertices are the same, but for each arc $a$ in $A$, there are $M$ copies

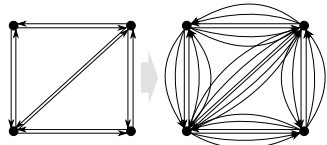

Figure 9: Evolution from $G$ to expanded $G^{\text{exp}}$.

$(\tilde{a}_m)_{m\in\{1,\dots,M\}}$ of $a$ in $A^{\text{exp}}$ (see Figure 9). We then define the capacity $u_{\tilde{a},m} = \tau_m - \tau_{m-1}$. Note that we can only consider equal capacities $u_{\tilde{a},m} = u_{\tilde{a}} \ \forall m \in \{1,\dots M\}$. The equivalence in the non-regularized case is straightforward. For example, in a setting with a capacity $u_{\tilde{a},m} = 3$: The first three agents traversing arc $a$ experience the cost $\theta_{a,1}$, the second three agents traversing arc $a$ experience the cost $\theta_{a,2}$, etc. We can then use the two previously introduced regularizations. Squared Euclidean regularization now involves an orthogonal projection of $\theta$ on $\bar{\mathcal{Y}}_{\boldsymbol{u}}^{\text{exp}}$, while Monte-Carlo approximations in the perturbation case involve solving a maximum cost flow on $\bar{\mathcal{Y}}_{\boldsymbol{u}}^{\text{exp}}$.

### C.4 BACKGROUND ON POLYNOMIAL LATENCIES REGULARIZED BY PERTURBATION

Lastly, we introduce monotonously increasing latency functions $\ell_a^{\boldsymbol{\theta}}(\bar{y}_a)$ that depend on the respective traffic flow $\bar{y}_a$. To do so, we introduce a feature mapping $\boldsymbol{\sigma}$ that maps any component $y \in \mathbb{R}_+$ to a feature vector $\boldsymbol{\sigma}(y) = (\sigma_k(y))_{k\in\{k,\dots,K\}} \in \mathbb{R}^K$ such that $y \mapsto \sigma_k(y)$ is convex for each $k$,

$$\boldsymbol{\sigma} : \boldsymbol{y} \in \mathcal{Y} \mapsto \boldsymbol{\sigma}(\boldsymbol{y}) \in \mathbb{R}^{A\times K}.$$

Then, we consider polynomial latency functions generated by the mapping $(\sigma_k)_k$,

$$\ell_a^{\theta}(\bar{y}_a) = \sum_k \theta_{k,a}\sigma_k(\bar{y}_a) \quad \text{with} \quad \theta_{k,a} \geq 0 \quad \text{for all } k \text{ and } a. \tag{32}$$

Here $\theta_{k,a}$ is the weight coefficient of the $k$-th basis function in $\ell_a$, and $\sigma_k$ is the respective basis function. The overall equilibrium is therefore parametrized by $\boldsymbol{\theta} = (\theta_{k,a})_{k\in\{1,\dots,K\}}^{a\in A}$ with $k$ indexing the mapping $(\sigma_k)_k$. Clearly, in this context, choosing the right family of function $\boldsymbol{\sigma}$ is a crucial design decision: it should be simple enough to lead to tractable WEs for any $\boldsymbol{\theta} \geq 0$, and at the same time sufficiently expressive to allow for good approximations.

In this work, we restrict ourselves to the polynomial family $\boldsymbol{\sigma}$ using $k = \{0,1\}$, and $\sigma_k(\bar{y}) = \bar{y}^k$. Then, the latency function for arc $a$ becomes

$$\ell_a^{\theta}(\bar{y}_a) = \theta_{0,a} + \theta_{1,a} * \bar{y}_a,$$

with $\bar{y}_a$ denoting the aggregated flow on arc $a$. We can interpret this latency function as follows: the intercept $\theta_{0,a}$ represents the traffic-free latency, and the slope $\theta_{1,a}$ represents the induced latency per agent on arc $a$. In general, we can use arbitrary complex polynomials to describe the latency function. The resulting latency function 32 is a convex function, as the basis functions $(\sigma_k)_k$ are strictly convex for each $k$.

We can extend the regularization by perturbation to that setting. Let us define $\mathcal{C} = \text{conv}(\sigma(\bar{\mathcal{Y}}))$. Remark that $\mathcal{C}$ is closed and convex but no more a polyhedron. We can now define the generalizations

$$F(\boldsymbol{\theta}) = \mathbb{E}\Big[\max_{\boldsymbol{\mu}\in\mathcal{C}}(\boldsymbol{\theta}+Z)^\top\boldsymbol{\mu}\Big] = \mathbb{E}\Big[\max_{\boldsymbol{y}\in\bar{\mathcal{Y}}}(\boldsymbol{\theta}+Z)^\top\boldsymbol{\sigma}(\boldsymbol{y})\Big], \quad \text{and} \quad \Omega(\boldsymbol{\mu}) = F^*.$$

We still have $F = \Omega^*$. The prediction is however now on $\mathcal{C}$

$$\hat{\boldsymbol{\mu}}_\Omega(\boldsymbol{\theta}) = \mathbb{E}\Big[\arg\max_{\boldsymbol{\mu}\in\mathcal{C}}(\boldsymbol{\theta}+Z)^\top\boldsymbol{\mu}\Big] = \arg\max_{\boldsymbol{\mu}\in\mathcal{C}}\boldsymbol{\theta}^\top\boldsymbol{\mu} - \Omega(\boldsymbol{\mu}) \neq \arg\max_{\boldsymbol{y}\in\bar{\mathcal{Y}}}\boldsymbol{\theta}^\top\boldsymbol{\sigma}(\boldsymbol{y}) - \Omega(\boldsymbol{\sigma}(\boldsymbol{y})). \tag{33}$$

We retrieve a prediction $\hat{\bar{\boldsymbol{y}}}$ by taking the projection on $\bar{\mathcal{Y}}$

$$\hat{\bar{\boldsymbol{y}}}(\boldsymbol{\theta}) = \mathbb{E}\Big[\arg\max_{\boldsymbol{y}\in\bar{\mathcal{Y}}}(\boldsymbol{\theta}+Z)^\top\boldsymbol{\sigma}(\boldsymbol{y})\Big].$$

It leads to the learning problem and Fenchel-Young loss

$$\min_{\boldsymbol{w}} \frac{1}{N}\sum_{i=1}^N \mathcal{L}_\Omega\big(\varphi_w(\boldsymbol{x}_i),\sigma(\bar{\boldsymbol{y}}_i)\big) \quad \text{where} \quad \mathcal{L}_\Omega(\boldsymbol{\theta},\bar{\boldsymbol{\mu}}) = \Omega^*(\boldsymbol{\theta}) + \Omega(\bar{\boldsymbol{\mu}}) - \boldsymbol{\theta}^\top\boldsymbol{\mu}.$$

Again, we can obtain Monte-Carlo evaluation of the gradient

$$\nabla_{\boldsymbol{\theta}}\mathcal{L}_\Omega(\boldsymbol{\theta},\bar{\boldsymbol{\mu}}) = \mathbb{E}\Big[\arg\max_{\boldsymbol{\mu}\in\mathcal{C}}(\boldsymbol{\theta}+Z)^\top\boldsymbol{\mu}\Big] - \bar{\boldsymbol{\mu}} = \mathbb{E}\Big[\arg\max_{\boldsymbol{y}\in\bar{\mathcal{Y}}}(\boldsymbol{\theta}+Z)^\top\sigma(\boldsymbol{y})\Big] - \bar{\boldsymbol{\mu}}$$

which is quite convenient as we can make the computations on $\bar{\mathcal{Y}}$. Using a squared Euclidean regularization is not as convenient computationally as we would need to optimize over $\boldsymbol{\mu}$ instead of $\bar{\boldsymbol{y}}$ due to the inequality in Equation 33.

## D    TRAFFIC SCENARIOS

We consider four stylized and two realistic scenarios for our experiments. While the stylized scenarios allow us to isolate effects that highlight the differences between our COAML pipelines and pure ML approaches, the realistic scenarios allow us to highlight the efficiency of our COAML pipelines in practice.

The stylized scenarios consider randomly generated networks using the model from Eisenstat (2011) similar to Figures 2a, 2b, and 2c. All roads in these networks have the same characteristics with respect to capacity. We further consider 100 home and 100 work locations in the network and a set of 200 planned trips in the form of origin-destination pairs. Precisely, we consider 100 planned trips starting at the home locations and driving to the work locations and 100 planned trips starting at the work locations and driving to the home locations.

In the first two scenarios, we derive the target traffic equilibrium via solving a time-expanded WE with 20 discrete time steps. The WE considers a latency function $\ell_a(\bar{y}_a) = d_a + \bar{y}_a$ with $d_a$ representing the length of arc $a$ and $\bar{y}_a$ denoting the aggregated flow on arc $a$.

**High-entropy (HE) scenario:**    The *high-entropy scenario* comprises instances that each consist of a randomly generated network and uniformly distributed home and work locations across the respective network (see Figure 2a). This uniform distribution of home and work locations leads to an evenly distributed traffic flow with increasing traffic flow in the center of the network and a lower traffic flow in the outer area of the network. This scenario provides a traffic equilibrium that highly correlates with the context of the network. Precisely, the traffic flow of an arc correlates with the location of the respective arc in the network. Thus, context features like the coordinates of an arc and the location of the arc within the network have a high correlation with the traffic flow. From a theoretical perspective, this scenario is good for learning traffic equilibrium prediction with pure ML approaches: the high correlation between context features and the target traffic flow allows us to learn a direct mapping from features to traffic flow independent of the combinatorial nature of traffic flows.

**Low-entropy (LE) scenario:**    In the *low-entropy scenario* each instance consists of a randomly generated network with home locations located in the upper right corner and work locations in the bottom left corner (Figure 2b). In such a scenario, the resulting traffic flow strongly depends on the combinatorial trip information: Trips start at the home locations in the upper right corner and end at the work locations in the bottom left corner and vice versa. In comparison to the high-entropy scenario, the traffic equilibrium resulting in the low-entropy scenario is less context-sensitive but more sensitive to the combinatorial relation between home and work locations. From a theoretical perspective, pure ML approaches are not expected to yield good solutions in this scenario.

For the remaining scenarios, we derive the target traffic equilibrium by running the agent-based transport simulation MATSim. We refer to Appendix D.1.2 for detailed information on the MATSim simulation.

**Square-world (SW) scenario:**    The *square-world scenario* utilizes the road network of the stylized scenario, but the origin and destination locations are distributed according to the distribution of roads in the respective network. We simulate a one-hour epoch, with all agents starting their trips from their home locations to their work locations at minute zero to mimic the morning rush hour, and their trips from their work locations to their home locations after 30 minutes to mimic the evening rush hour. This scenario complements our set of stylized scenarios: it does not mimic a real-world setting but allows to generate insights on the performance of COAML pipelines in comparison to pure ML pipelines when it comes to predicting combinatorial information.

**Berlin scenario:**    The *Berlin scenario* comprises instances that consist of district networks (Figure 2d) extracted from a realistic road network of the city of Berlin. The origin-destination pairs are extracted from a calibrated real-world MATSim scenario.[1] If a trip passes the district but has an

---

[1]https://github.com/matsim-scenarios/matsim-berlin?tab=readme-ov-file

origin/destination outside of the district, we set the origin/destination of the respective trip to the border of the district. Thus, this scenario serves to generate insights into how well our benchmarks can predict traffic equilibria for certain districts when contextual information is only available for the respective district, but the contextual information for the outer area of the district is unavailable, e.g., investigating a traffic management effect on a certain district, when census data of the outer area of the district is unavailable.

**Berlin artificial population (Berlin-AP) scenario:** The *Berlin artificial population scenario* comprises instances that consist of district networks extracted from a realistic road network of the city of Berlin, but the population plans are artificially generated so that all trips start and end within the district network and there are no trips passing through the district network (Figure 2e). The home and work locations are distributed according to the density of the street networks, with more home and work locations in areas with a dense arc network and fewer home and work locations in areas with a sparse arc network.

**Square-world time-variant (SW-TV) scenario:** In the aforementioned scenarios, we only focused on the aggregated traffic flow over time. However, in many traffic applications, traffic flow needs to be predicted over time. Therefore, we consider a *square-world time-variant scenario*, which investigates the performance of our benchmarks when predicting traffic flow over time. The challenge is that MATSim yields a traffic flow over constant time, but our benchmarks predict a traffic flow over discrete time. To circumvent this problem, we discretize the target traffic flow derived from the MATSim simulation over time. To do so, we apply a time expansion on the transport networks $G$ of the *square-world scenario* and measure the MAE on the time-expanded transport networks. Specifically, a time-expansion on the transport network $G$ refers to discretizing time into disjoint epochs $T$; then each arc $a \in A^{|A| \times |T|}$ in the time-expanded network represents an arc with a specific time index $t \in T$. On the spatial dimension, the length of an arc specifies the spatial distance from the start location to the end location of the arc, and from a temporal perspective, the length of an arc specifies the time it takes to traverse the respective arc. Note that in such a setting, the latency on arc $a$ at time $t_1 \in T$ does only depend on the time-variant traffic flow $\bar{y}_{a,t_1}$ on the respective arc, but not from the flow $y_{a,t_2}$ at another point in time $t_2 \in T$. Intuitively, we assume that the traffic flow on an arc is independent of previous traffic flows on the respective arc.

For all scenarios, we create 9 training instances, 5 validation instances, and 6 test instances. We run the training for a maximum of 20 hours, or 100 training epochs. We run the experiments on a computing cluster using 28-way Haswell-EP nodes with Infiniband FDR14 interconnect and 2 hardware threads per physical core. We report the MAE on the test instances.

### D.1    WARDROP EQUILIBRIUM SOLVERS

In general, WE solvers seek a set of flows in $\mathcal{Y}$ such that each agent $j$ travels on a path in $\mathcal{Y}^{(j)}$ where unilaterally deviating from this path would always incur a longer travel time for the agent. Correa & Stier-Moses (2011) provide a good intuition for WEs and respective solvers.

In general, we can divide WE solvers into analytical solvers and simulation-based solvers. Analytical solvers require knowledge about the system's latency functions to allow the computation of the respective equilibria, which is why they are frequently used for stylized or synthetic analyses. In practice, these latency functions are usually not known and consequently simulation-based solvers are used to determine the respective equilibria. The drawback of such simulation-based solvers is their computational time, which can easily exceed several hours depending on the size of the case study and the hyperparameters chosen.

Our approach aim to bridge the gap between the accuracy of simulation-basd solvers and the tractability of analytical solvers by learning the parameterization of the respective latency functions and embedding them into a CO-layer. In the following, we give an overview of related analytical and simulation-based solvers.

D.1.1  ANALYTICAL SOLVERS.

Analytical solvers leverage mathematical formulations and iterative approaches. The analytical solvers mainly base on two intuitions:

1. The Beckmann formulation (Beckman et al., 1956) interprets the WE problem as a multi-commodity flow problem (MCFP) and states that the convex optimization problem

$$\min_{y \in \bar{\mathcal{Y}}} Z(y) = \sum_{a \in A} \int_0^{y_a} \ell_a(x) dx \tag{34}$$

that satisfies the *flow conservation* and the *non-negativity* of flows yields a WE.

2. We can formulate the notion of a WE as a variational inequality (Smith, 1979). Then a flow $y$ is a WE when

$$\sum_{a \in A} \ell_a(\bar{y}_a)\bar{y}_a \leq \sum_{a \in A} \ell_a(\bar{y}_a)x_a, \quad \forall x \in \bar{\mathcal{Y}} \tag{35}$$

holds.

In the following, we provide an overview over the most common analytical WE solvers.

**Frank-Wolfe Algorithm**  The Frank-Wolfe algorithm is an iterative approach solving Problem 34. Intuitively, the Frank-Wolfe algorithm starts with an initial flow, updates the travel times depending on the flow, and derives an direction to update the flow such that it reduces $Z(y)$. Iteratively, the algorithm updates the traffic flow $y$ to finally determine a WE. Formally, we detail this process in Algorithm 1.

---

**Algorithm 1** Frank-Wolfe Algorithm

---

**Require:** $\bar{y}^{(0)}$          ▷ Set initial flow
  $k \leftarrow 0$
  **while** WE not found **do**
    $x \leftarrow \min_x \sum_{a \in A} \ell_a(\bar{y}_a^{(k)}) * x_a$      ▷ Get direction to update $\bar{y}^{(k)}$
    $\alpha^{(k)} \leftarrow \arg\min_\alpha Z(\bar{y}^{(k)} + \alpha * (x - \bar{y}^{(k)}))$      ▷ Line search
    $\bar{y}^{(k+1)} \leftarrow \bar{y}^{(k)} + \alpha^{(k)} * (x - \bar{y}^{(k)})$      ▷ Update flow
    $k \leftarrow k + 1$
  **end while**

---

**Successive Averages Algorithm**  The Successive Averages algorithm is an heuristic iterative algorithm similar to the Frank-Wolfe Algorithm but different in the flow update process. We depict the process in Algorithm 2.

---

**Algorithm 2** Successive Averages Algorithm

---

**Require:** $\bar{y}^{(0)}$          ▷ Set initial flow
  $k \leftarrow 0$
  **while** WE not found **do**
    $x \leftarrow \min_x \sum_{a \in A} \ell_a(\bar{y}_a^{(k)}) * x_a$      ▷ Get direction to update $\bar{y}^{(k)}$
    $\bar{y}^{(k+1)} \leftarrow \bar{y}^{(k)} + \frac{1}{k+1} * (x - \bar{y}^{(k)})$      ▷ Update flow
    $k \leftarrow k + 1$
  **end while**

---

**Decomposition Algorithms**  Decomposition algorithms decompose the problems into subproblems in the sense that they separate flows by node of origin. The best known algorithm is the Bar-Gera's algorithm (Bar-Gera, 2002) that we show in Algorithm 3.

---

**Algorithm 3** Bar-Gera's Algorithm

---

**Require:** $\bar{y}^{(0)}$                                             ▷ Set initial flow
  $k \leftarrow 0$
  **while** WE not found **do**
    **for** $o \in V$ **do**                             ▷ Iterate over all origins in the network
      $\bar{y}^o \leftarrow \min_{\bar{y}^o \in \bar{\mathcal{Y}}^o} \sum_{a \in A} \ell_a(\bar{y}_a^{(k)}) * \bar{y}_a^o$        ▷ Distribute flows from origin $o$
    **end for**
    $\bar{y}^{(k+1)} \leftarrow \cup_{o \in V} \bar{y}^o$                    ▷ Combine flows from all origins
    $k \leftarrow k + 1$
  **end while**

---

**Projection Methods**     Projection methods leverage the variational intequalities 35 and project flows onto feasible flow sets. We can reformulate the variational inequalities as an optimization problem,

$$\text{find } y \in \mathcal{Y} \text{ such that } \langle \ell(\bar{y}), \boldsymbol{x} - \bar{\boldsymbol{y}} \rangle \geq 0, \quad \forall \boldsymbol{x} \in \bar{\mathcal{Y}} \tag{36}$$

then the objective is to find a traffic flow $\boldsymbol{y}$ that satisfies the variational inequalities, and we receive this traffic flow with the following Algorithm 4.

---

**Algorithm 4** Projection Methods

---

**Require:** $\bar{y}^{(0)}$                                             ▷ Set initial flow
  $k \leftarrow 0$
  **while** WE not found **do**
    $\boldsymbol{x} \leftarrow \bar{y}^{(k)} - \lambda \ell(\bar{y}^{(k)})$                  ▷ Get new flow proposition
    $\bar{y}^{(k+1)} \leftarrow \mathcal{P}_{\bar{\mathcal{Y}}}(\boldsymbol{x})$      ▷ Project flow proposition onto the feasible flow set $\bar{\mathcal{Y}}$
    $k \leftarrow k + 1$
  **end while**

---

**Mathematical programming**     Depending on the on form of the latency function $\ell$ we can apply various mathematical programming methods like, e.g., Barrier Methods or Interior Point Methods to find a WE via solving the Beckman formulation, Problem 34.

### D.1.2   Simulation-based solvers: MATSim

In comparison to analytical solvers we can also leverage agent-based simulations that simulate the actions of each agent individually, and by following a selfish behaviour of each agent yielding a WE. Instead of presenting various different simulation approaches, we present MATSim (Horni et al., 2016), as we also consider MATSim in Section 5 to yield target traffic flows $\bar{y}^*$.

MATSim is an agent-based transport simulation. Specifically, each agent in the simulation follows an agent-specific plan. This agent plan details the agent's daily trips, including starting times and the respective routes. The simulation follows a queue-based approach such that when a vehicle enters a network link from an intersection, the simulation adds the vehicle to the tail of vehicles waiting to traverse the link (Vickrey, 1969). The waiting time depends on storage capacity and the flow capacity of the respective link. The resulting traffic flow $\bar{y}^*$ results from a co-evolutionary optimization approach with each agent representing a single species, meaning that each agent optimizes its respective plan individually. Specifically, in each evolution step, each agent considers a plan. Then, the simulation simulates all plans in the system and distributes scores for each individual plan depending on waiting times and deviations from initial schedules. Subsequently, the co-evolutionary approach updates each agent's plan via inheriting from previous plans from the same agent, mutation, and recombination actions. Thus, this co-evolutionary process improves each agent plan individually, dependent on all other plans in the system. Finally, this co-evolutionary approach leads to a stochastic user equilibrium $\bar{y}^*$. In this equilibrium, no agent can unilaterally improve its plan.

## E    WARDROPNET PIPELINES AND ML BASELINES

This section details all the benchmarks that we consider in our numerical study. While the first two benchmarks are pure deep learning benchmarks, the last three benchmarks relate to the COAML pipelines introduced in Section 4.

**FNN:** we consider a pure ML pipeline $f(\boldsymbol{x}) = \bar{\boldsymbol{y}}$ with an FNN model that was trained via supervised learning. To this end, we consider the FNN model and the related features presented in Appendix $B$.

**GNN:** we consider a pure ML pipeline $f(\boldsymbol{x}) = \bar{\boldsymbol{y}}$ with a GNN model that was trained via supervised learning. To this end, we consider the GNN model and the related features presented in Appendix $B$.

**Constant latencies (CL):** we consider a COAML pipeline using *constant latencies regularized by perturbation*. The COAML pipeline solves a MCFP with piecewise decomposed latency functions. We enable the piecewise decomposition by applying the network expansion trick as detailed in Appendix C.3: we multiply each arc $M$-times and set arc-specific capacities on each arc. We further consider a statistical model $\varphi_{\boldsymbol{w}}$ similar to the FNN model used in the *FNN* benchmark. We only add a last layer with a *softplus* function which additionally negates the output. Recall that we need to enforce the statistical model to predict negative latencies as we solve the equilibrium problem, which normally minimizes the latencies for all agents, as a maximization problem (cf. Equations 5-7) to allow the formulation of the Fenchel-Young loss (cf. Equation 8).

**Polynomial latencies (PL):** we consider a COAML pipeline using *polynomial latencies regularized by perturbation*. The COAML pipeline solves a WE. We assume a polynomial latency function with $k = \{0, 1\}$, and $\sigma_k(y) = y^k$. In this setting, we consider two branches of FNNs $(\varphi_{\boldsymbol{w}_0}, \varphi_{\boldsymbol{w}_1})$ to predict the intercept $\boldsymbol{\theta}_0$ and the slope parameter $\boldsymbol{\theta}_1$ of the latency function. Each FNN branch equals the FNN model used in the *FNN* benchmark.

**Euclidean regularization (ER):** we consider a COAML pipeline using *latencies with Euclidean regularization*. The COAML pipeline solves a WE. We assume a polynomial regularization term $\Omega(\boldsymbol{y}) = \frac{1}{2}||\bar{\boldsymbol{y}}||^2$. This enables us to learn the pipeline considering the WE as a regularized minimum cost flow problem. The statistical model is a FNN model similar to the statistical model considered in the *FNN* benchmark.

**Discussion:** The pure ML benchmarks, namely the *FNN* and the *GNN* benchmarks are not state-of-the-art deep learning benchmarks for predicting traffic equilibria. However, using the same deep learning models in the pure ML benchmarks and the COAML pipelines, namely the *CL*, *PL*, and *ER* benchmarks, allows meaningful comparison with respect to the impact of structured learning in comparison to purely supervised learning. A comprehensive tuning of deep learning benchmarks would lead to a standalone paper that focuses on deep learning architectures in the realm of traffic equilibrium prediction. However, the focus of this paper is to introduce the effectiveness of combining ML and CO in COAML pipelines.

## F    IN-DEPTH RESULTS DISCUSSION

This section presents our numerical results. In this section, we first focus on the results of the COAML pipelines and the pure ML pipelines on the stylized and realistic scenarios. Second, we present a structural analysis to generate insights on the contribution of the benchmarks by visualizing the predictions of the COAML pipelines and the pure ML pipelines. Finally, we present the results of the benchmarks for the time-variant predictions.

**Stylized scenarios**    Figure 10a shows the results for the various benchmarks on the *high-entropy scenario*. All the COAML benchmarks, namely *CL*, *PL*, and *ER* outperform the pure ML benchmarks *FNN* and *GNN*. Precisely, the *CL* benchmark outperforms the pure ML benchmarks by around 60% on average, while the *PL* benchmark outperforms the *ML* benchmark by around 52% on average. Recall that predicting the traffic equilibrium in such a scenario depends less on the combinatorial trip information but rather on the location of an arc in the network, as the traffic flow is evenly distributed with increasing traffic flow to the center. Therefore, from a theoretical perspective, the

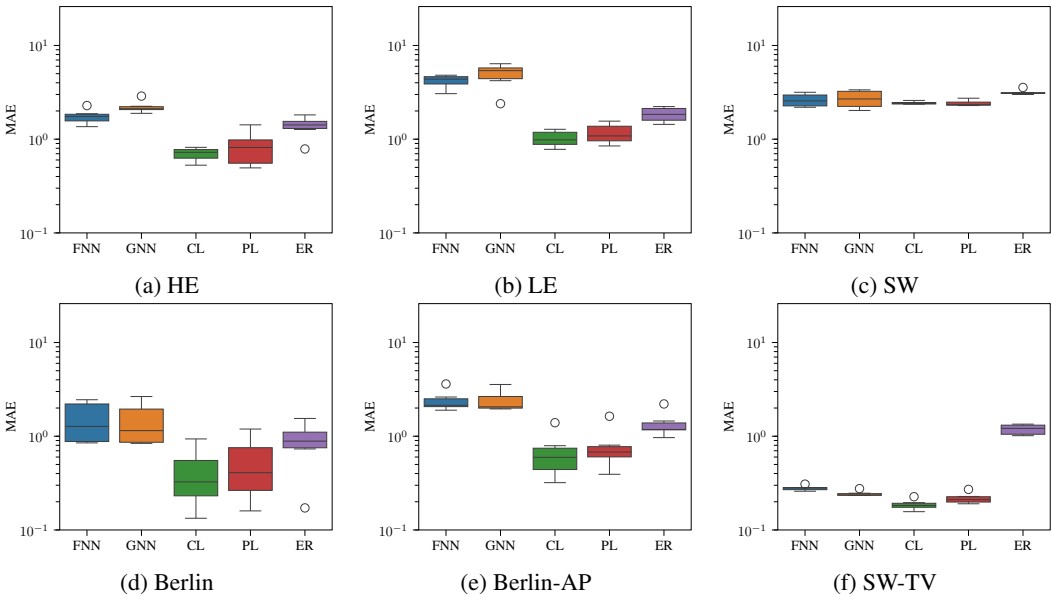

Figure 10: Benchmark performances on various stylized and realistic traffic scenarios.

pure ML pipeline should yield good results in this scenario, as there is a high correlation between the traffic flow and the context features describing the location of an arc within the network. The results show the superiority of COAML pipelines over pure ML pipelines, even in scenarios that exhibit a structure that is easy to capture with pure ML pipelines.

Figure 10b shows the results on the *low-entropy scenario*. The MAE for the *FNN* benchmark increased from around 1.75 in the *high-entropy scenario* to around 4.18 in the *low-entropy scenario*. This indicates that the *low-entropy scenario* is indeed more challenging to predict for pure *ML* pipelines in comparison to the *high-entropy scenario* as the correlation between the arc flow and the context features is lower. Comparing the different benchmarks, we observe that the COAML pipelines again outperform the pure ML pipelines. Remarkably, the prediction error of the COAML pipelines in the *low-entropy scenario* equals the prediction error in the *high-entropy scenario*. Specifically, the *CL* benchmark decreases the prediction error of the *FNN* benchmark by around 75% on average, and the *PL* benchmark decreases the prediction error of the *FNN* benchmark by around 72% on average. The experiments on the *high-entropy scenarios* and the *low-entropy scenarios* indicate that the COAML pipeline learns to leverage the statistical model and the equilibrium layer depending on the underlying task outperforming pure ML pipelines, even in settings when pure ML pipelines are expected to perform well: while the contribution of the statistical model is particularly important in the *high-entropy scenario* when it comes to extracting the right context features, the equilibrium layer is particularly important in the *low-entropy scenario* when it comes to processing combinatorial information. Surprisingly, the *GNN* benchmark does not perform well in both scenarios. This can be attributed to two reasons: first, while the GNN models perform particularly well when predicting temporal traffic forecasts in a receding horizon manner, the GNN model might be less well-suited when predicting aggregated traffic flows. Second, in comparison to an FNN model, the GNN model accounts for irregular structures with many intertwined dependencies of neighboring nodes and therefore is prone to over-smoothing. Thus, a simple GNN model might not be able to extract notable features.

Figure 10c shows the results for the *square-world scenario*. Here, all approaches outperform the *ER* benchmark. Still, the *CL* benchmark and the *PL* benchmark outperform the *FNN* benchmark by around 7% on average. The pure ML pipelines reach a similar absolute performance as in the *high-entropy scenario*, and the COAML pipelines reach a lower absolute performance in comparison to the *high-entropy scenario* and the *low-entropy scenario*. Here, a similar effect arises as in the *high-entropy scenario*: Roads located in a higher populated area have a higher traffic flow. Thus, there is a correlation between the arc context, namely the amount of work and home locations in the closer neighborhood, and the resulting traffic flow. So, from a theoretical perspective, the pure

ML benchmarks should yield good results in this scenario. However, the COAML benchmarks still outperform the pure ML benchmarks.

**Realistic scenarios**    Figure 10d shows the results for the *Berlin scenario*. Here, the COAML pipelines *CL* and *PL* significantly outperform the pure *ML* pipelines. Precisely, the *CL* benchmark reduces the prediction error by around 72% on average in comparison to the *FNN* benchmark, and the *PL* benchmark reduces the prediction error by around 64% on average. Note that the results are biased due to many trips starting and ending at the border of the network. Intuitively, the trips starting and ending at the border of the network are long trips passing through the Berlin district network. Thus, most of these trips focus on large roads with large capacity and free speed, mainly highways. From the pure ML perspective, it is difficult to learn the focus of these through-passing trips on single roads, i.e., roads forming the shortest path from one side of the district to the other side of the district, and the high difference in traffic flow between these shortest-path-forming roads and feeder roads. Moreover, the origin locations and destination locations of these trips depend on the outer context that is not included in the scenario context, making it even more difficult for pure ML benchmarks to learn the relationship between the scenario context at hand and the focus of the traffic on single main roads. On the other side, the COAML benchmarks can leverage the equilibrium layer to learn the combinatorial effects of many trips, focusing on the shortest-path-forming roads.

Figure 10e shows the results of the *Berlin artificial population scenario*. Recall the scenario equals the *Berlin scenario*, but instead of the calibrated Berlin population, the scenario considers an artificially generated population restricted to the respective districts. Still, the results are similar to the results for the *Berlin scenario*: The *CL* and *PL* benchmarks outperform all pure ML benchmarks. Precisely, the *CL* and the *PL* benchmarks outperform the *FNN* benchmark by around 71% and 67% on average, respectively. Intuitively, we observe a similar effect as in the *Berlin scenario* with the traffic flow focusing on the main roads forming shortest paths between areas with high densities of work and home locations. While pure ML benchmarks fail to account for the combinatorial nature of the resulting traffic flow, the COAML pipelines leverage the statistical model and the equilibrium layer to combine the processing of contextual information with the combinatorial nature of traffic equilibria focusing on high-volume roads. Surprisingly, the *CL* benchmark outperforms the *PL* benchmark in all scenarios, although the *CL* benchmark solves a MCFP in the CO-layer and the *PL* solves a WE in the CO-layer. Originally, the *CL* benchmark was introduced as an efficient approximation for COAML pipelines solving a complex WE in the CO-layer. However, the superior performance of the *CL* benchmark might have a natural reason: According to Vickrey (1969), we can model flows with a queueing approach – also MATSim leverages such a queueing approach – such that latencies of an arc only increase when its respective queue reaches a certain length. We can interpret this stepwise increase of latencies depending on the number of agents in a queue as a piecewise latency function, similar to how we model it in the *CL*-pipeline. This observation would lead to the assumption, that stepwise latency functions as used in the *CL*-pipeline lead to better traffic equilibrium approximations as polynomial latency functions as used in the *PL*-pipeline.

Overall, we can conclude that the COAML benchmarks outperform the pure ML benchmarks on all tested scenarios, also in scenarios that in particular serve to yield good predictions for pure ML benchmarks. In Appendix F.1, we introduce further stylized experiments and report respective results.

**Structural analysis:**    In this paragraph, we provide a structural analysis to substantiate the intuition on why the COAML benchmarks outperform pure ML benchmarks in predicting traffic equilibria. Figure 11 shows the target (11a) and predicted (*FNN*: 11b; *GNN*: 11c; *CL*: 11d; *PL*: 11e; *ER*: 11f) traffic flow aggregated over time for the various benchmarks on the *Berlin scenario* (Figure 2d). Note that in Figure 11a the main traffic flow focuses on the main roads, i.e., roads forming the shortest paths between areas of interest, while there is only a low flow on small roads, i.e., local roads in residential areas. As can be seen, the pure *ML* benchmarks (Figures 11b and 11c) focus on predicting good mean values for the traffic flow but fail to map realistic traffic flow patterns that reveal high traffic flows on main roads. Analyzing the difference between the pure ML benchmarks, the *GNN* benchmark yields a more realistic traffic flow pattern than the *FNN* benchmark. This observation is in line with the different architectures of the *FNN* benchmark and the *GNN* benchmark: the *GNN* benchmark allows the processing of contextual information and embeddings from neighboring roads and enables the detection of main roads to represent consistent flow attributes. Contrarily, the

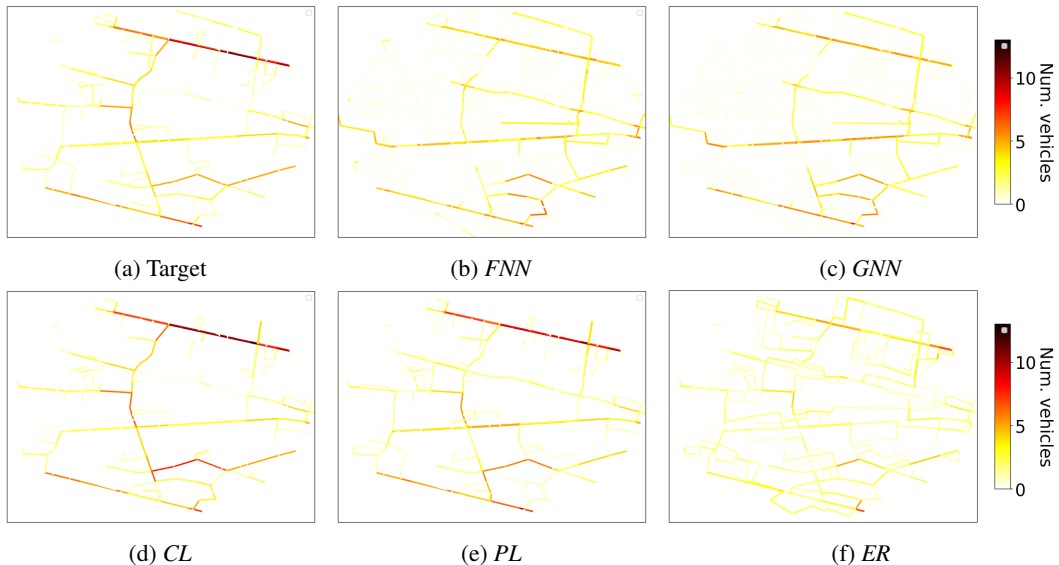

Figure 11: Visualization of time-invariant traffic flows.

*FNN* benchmark only considers an arc-specific context and thus yields uncorrelated traffic predictions between roads (Figure 11b). Still, the *FNN* benchmark outperforms the *GNN* with respect to accuracy due to the better performance in predicting traffic flows on small roads. In contrast, the COAML benchmarks, namely the *CL* and *PL* benchmarks (Figure 11d and 11e) predict a realistic traffic equilibrium with a high traffic flow on main roads and a low traffic flow on small roads. The remaining *ER* benchmark, which is also a COAML pipeline, fails to mimic realistic traffic equilibria but focuses more on predicting small traffic flows on rarely used small roads. This might be due to the restricted latency function used in the *ER* benchmark that only allows us to learn the y-intercept, which is a strong limitation when it comes to predicting complex traffic patterns. Studying richer latency functions remains an interesting avenue for future work in this context.

Figure 12 shows the mapping from target traffic flow values to predicted traffic flow values and substantiates our conclusions drawn from analyzing Figure 11. Intuitively, in the case of a perfect correlation between the target and the predicted traffic flow, all points would be on the diagonal. However, focusing on the pure ML benchmarks, namely the *FNN* (Figure 12a) and *GNN* (Figure 12b) benchmarks their predictions have a low correlation with the target traffic flows especially for high traffic flow values on main roads and thus fail to represent a realistic traffic equilibrium pattern. In comparison, the COAML benchmarks, namely the *CL* (Figure 12c), the *PL* (Figure 12d), and the *ER* (Figure 12e) benchmarks yield predictions with a high correlation to the target traffic flows especially for high traffic flow values on the main roads.

Overall, our structural analyses show the limitations of pure ML pipelines compared to COAML pipelines: pure ML pipelines fail to learn the combinatorial dynamics of the underlying traffic patterns and thus yield predictions with a low correlation to the target traffic flow values, especially for high traffic flow values on the main roads. From a pure ML perspective, it is difficult to learn that the traffic flows often spread largely over a limited number of main roads. In contrast, the COAML

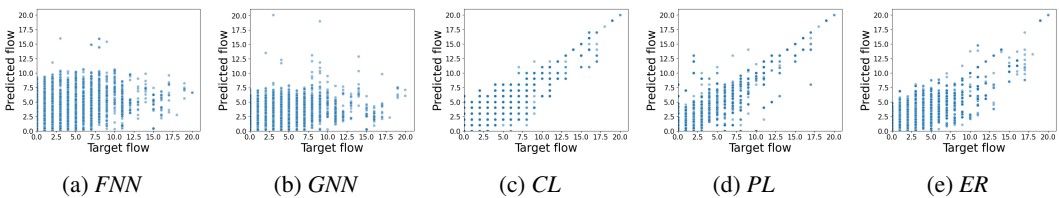

Figure 12: Comparison of target and predicted traffic flows per arc for the Berlin scenario.

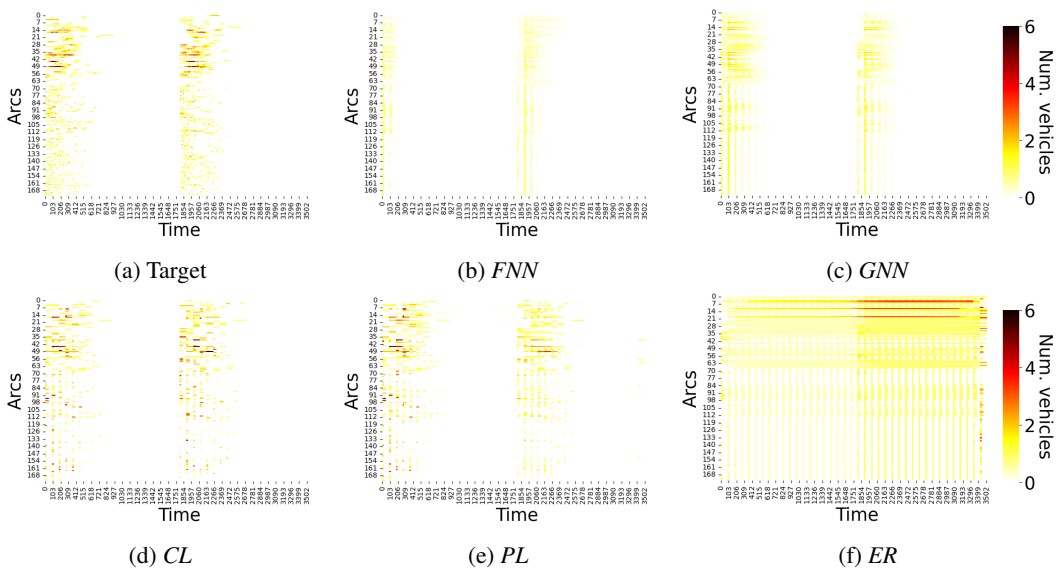

Figure 13: Visualization of time-variant traffic flows.

benchmarks leverage the equilibrium layer to learn the combinatorial nature of traffic equilibria and thus allow the prediction of realistic traffic patterns.

**Time-variant traffic flows:** Figure 10f shows the results of the *square-world time-variant scenario*. As can be seen, the *CL* and the *PL* benchmarks outperform all other benchmarks. More specifically, the *CL* benchmark reduces the prediction error in comparison to the *FNN* benchmark by around 33% on average and the *PL* benchmark reduces the prediction error in comparison to the *FNN* benchmark by around 22% on average. Note that in this setting, we did not learn piecewise decomposed latency functions but expanded the underlying transport network graph over time. The time expansion of the WE problem significantly increases the runtime for finding the WE in the CO-layer. To circumvent long training times, we ignored the perturbation during training as the aggregated traffic flow $\bar{y}$ acts similarly to a regularization term. However, the missing perturbation explains the slightly worse performance of the *PL* benchmark compared to the *CL* benchmark. In general, the MAE of the benchmarks is lower than in the other scenarios, which is due to the many zero values in the target traffic equilibrium over the time-expansion. Note that the *ER* benchmark fails to account for the time-expansion, because the benchmark only allows for learning the y-intercept, and embedding time-related information and spatial-related information within a single y-intercept remains difficult. Figure 13 visualizes the prediction of the traffic flow over time. We recall that agents schedule trips from home to work at minute zero, which reflects the morning rush hour, and from work to home after 30 minutes which reflects the evening rush hour. When comparing the target traffic flow values (Figure 13a) with the ML benchmarks (*FNN*: 13b, *GNN* 13), we see that the ML benchmarks underestimate the traffic flow and focus more on predicting a good mean value. The COAML benchmarks (*CL*: Figure 13d, *PL*: 13e) predict realistic traffic flow patterns in line with the target traffic flow (Figure 13). This insight is consistent with the observations made when predicting traffic flows aggregated over time (cf. Figure 11). The *ER* benchmark (Figure 13f) fails to account for the time component such that the traffic distributes equally over time. This effect might results from the simplified latency function used in the *ER* benchmark that only allows us to learn the y-intercept.

### F.1 COMPLEMENTARY STYLIZED EXPERIMENTS

In this section, we present benchmark performances for 16 different scenarios. Each scenario is a combination of an *environment* and an underlying *oracle*. While the *environment* details the underlying street network which is generated from the model of Eisenstat (2011) and the population with the respective agent plans, the *oracle* defines how to derive the target traffic equilibrium. In each

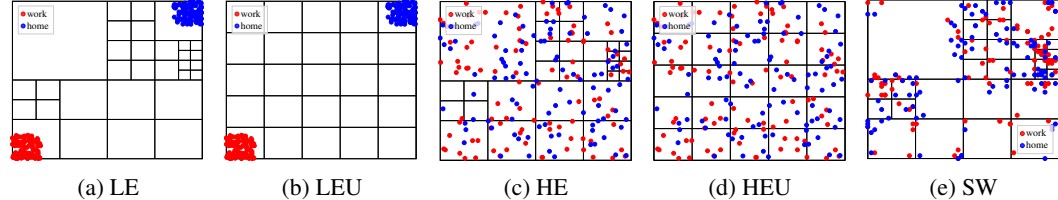

(a) LE  (b) LEU  (c) HE  (d) HEU  (e) SW

Figure 14: Illustration of the environments.

scenario, we consider 100 agents, and each agent has a home and work location. Each agent travels in the morning from its home location to its work location and in the evening from its work location to its home location. We solve all scenarios in a time-expanded setting with 20 discrete time steps.

We consider the *environments* shown in Figure 14:

**Low-entropy (LE):** A randomly generated street network. All home locations are in the upper right corner and all work locations are in the lower left corner. Each instance of this scenario considers a random street network and random home and work locations.

**Low-entropy uniform (LEU):** A grid street network. All home locations are in the upper right corner and all work locations are in the lower left corner. Each instance of this scenario considers random home and work locations.

**High-entropy (HE):** A randomly generated street network. All home locations and work locations are uniformly distributed. Each instance of this scenario considers a random street network and random home and work locations.

**High-entropy uniform (HEU):** A grid street network. All home locations and work locations are uniformly distributed. Each instance of this scenario considers random home and work locations.

**Square world (SW):** A randomly generated street network. All home locations and work locations are distributed according to the distribution of the roads. Each instance of this scenario considers a random street network and random home and work locations.

We consider the following *oracles*:

**EasyMCFP:** The oracle solves a MCFP. The costs of the MCFP equal the street length.

**EasyWE:** The oracle solves a WE. The latency function of the WE for arc $a$ is $d_a + \bar{y}_a$ with $d_a$ representing the length of arc $a$ and $\bar{y}_a$ denoting the aggregated flow on arc $a$.

**randomMCFP:** The oracle solves a MCFP. The costs of the MCFP are uniformly distributed $c_a \sim \mathcal{U}(0, 100)$.

**randomWE:** The oracle solves a WE. The latency function of the WE for arc $a$ is $\theta_{1,a} + \theta_{2,a} * \bar{y}_a$ with $\theta_{1,a} \sim \mathcal{U}(1, 100)$ randomly distributed, $\theta_{2,a} \sim \mathcal{U}(1, 20)$ randomly distributed, and $\bar{y}_a$ denoting the aggregated flow on arc $a$.

Figure 15 shows the performance of the benchmarks introduced in Appendix E over the introduced scenarios. In general, the COAML benchmarks, namely the *CL* and the *PL* benchmarks, outperform the pure ML benchmarks, namely the *FNN* and *GNN* in almost all scenarios. The *ER* benchmark has a mixed performance over the scenarios, which can be attributed to the simplified latency functions that only allows learning the y-intercept. In the following, we interpret the results depicted in Figure 15. We first focus on scenarios considering an *EasyMCFP* oracle to yield target traffic equilibria. In these scenarios, the resulting traffic flow only depends on the shortest paths in the network with respect to arc lengths, but no traffic effects are considered. Intuitively, the *CL* pipeline yields the best results in all scenarios. Here, the *CL* would mimic the oracle when it learns to focus only on the arc length in the statistical model. In general, over all scenarios considering an *EasyMCFP* oracle the differences in accuracy between the benchmarks are rather small. In the scenarios that consider an *EasyWE* oracle to yield target traffic equilibria, the differences in accuracy between the different benchmarks are larger. This is intuitive, as the oracle yields target traffic equilibria that depend on actual congestion states. Thus, these target traffic equilibria are difficult to predict with pure ML

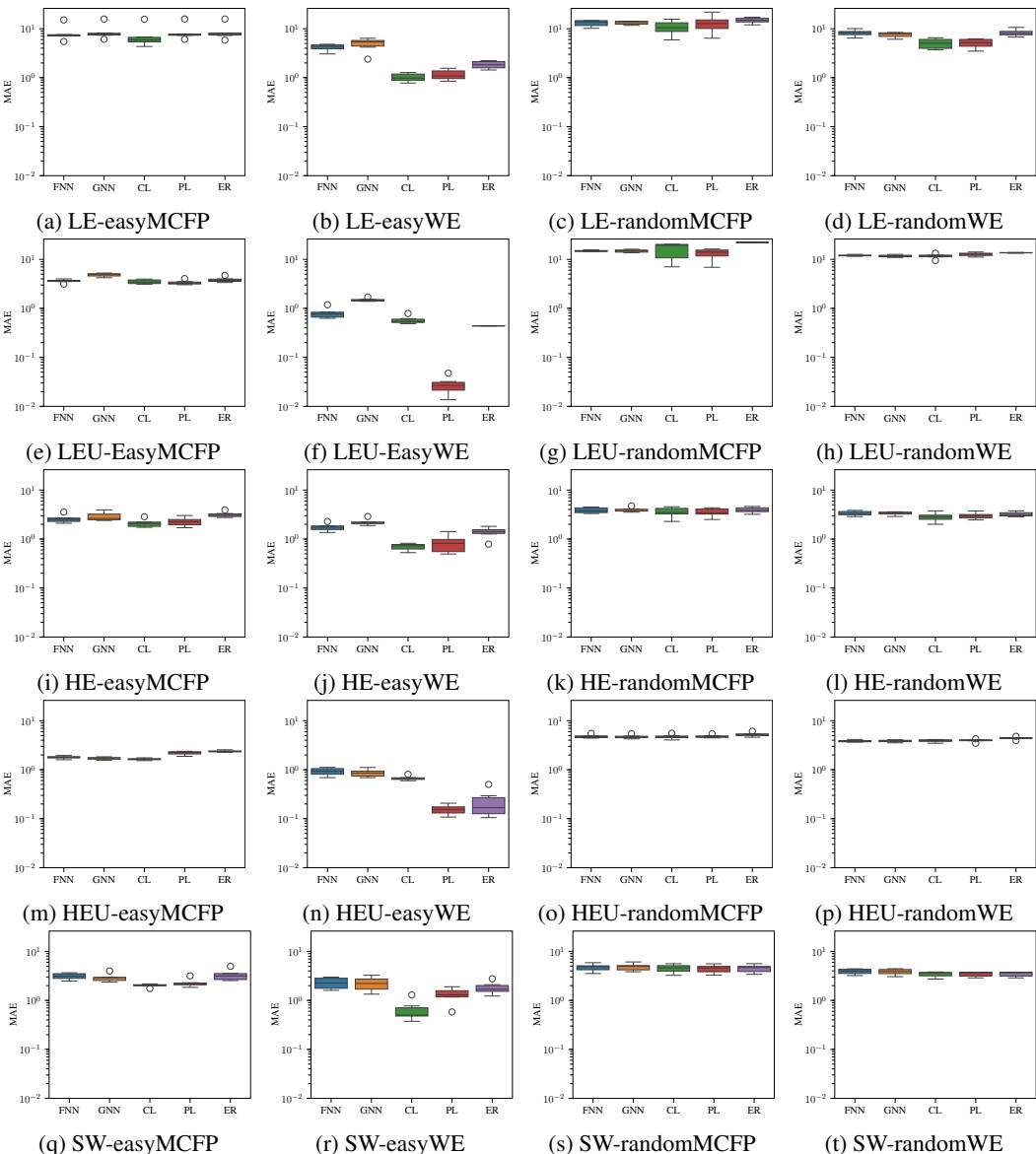

Figure 15: Performance of benchmarks on different scenarios.

benchmarks when only considering contextual information. In this setting, the COAML pipelines yield good results. Especially the *PL* and *CL* benchmarks perform well in all scenarios. While the *PL* benchmark performs particularly well in the *low-entropy uniform scenario* and the *high-entropy uniform scenario*, the *CL* benchmark performs particularly well in the *square-world scenario*. The pure ML benchmarks lead to the worst results over all scenarios. All the tests on scenarios with the *RandomMCFP* and the *RandomWE* oracle yield bad accuracy values over all benchmarks. This is intuitive as the target traffic equilibria follow a random structure that is completely independent of the underlying context. Thus, the prediction accuracies on these scenarios are worse than the prediction accuracies on the scenarios that are based on the *EasyMCFP* and the *EasyWE* oracle over all tested benchmarks. Although the benchmarks can not use any reasonable contextual information, the COAML benchmarks can leverage combinatorial information with respect to the origin and destination location of trips in the network and, therefore, slightly outperform the pure ML benchmarks for these scenarios.

