# OpenReview forum: "WardropNet: Traffic Flow Predictions via Equilibrium-Augmented Learning"
_ICLR.cc/2025/Conference — ICLR 2025 Poster_

### Official Review · Reviewer_qmEt · 2024-10-24

**Soundness:** 3
**Presentation:** 2
**Contribution:** 2
**Rating:** 5
**Confidence:** 4

**Summary:**

This paper proposes WardropNet to predict traffic flow by the combinatorial optimization-augmented machine learning (COAML) pipeline. Using supervised learning and Fenchel-Young loss, this method minimizes the difference between predicted and actual traffic flows by leveraging a Bregman divergence, ensuring it fits the geometry of traffic equilibria. WardropNet improves on average for traffic flow predictions compared with pure neural network method.

**Strengths:**

1. The essay provides a solid theoretical foundation for the approach.
2. The visualization and illustration are well made and help the readers to understand the paper.
3. By incorporating real-world scenarios, the method emphasizes its potential for real-world traffic management and urban planning.

**Weaknesses:**

1. Lack of comparison with more advanced baselines. While the paper compares WardropNet with basic machine learning models like FNN and GNN, it doesn't compare against other state-of-the-art traffic flow prediction models, such as deep reinforcement learning approaches or physics-informed neural networks.

2. Not well-structed essay presentation. As paper introduces a complex method, the explanation of how the model works is too technical too quickly. Meanwhile, the scenarios of the experiments section feels too long.

3. Sensitivity to Hyperparameters. The paper does not detail the sensitivity of WardropNet's performance to different hyperparameters such as layers number and learning rates. Since the model integrates multiple components, tuning could be more challenging, and it's unclear how stable the model is across different settings.

**Questions:**

1. In the end of section 1, the paper put up with " backpropagation through general, possibly combinatorial, equilibrium layers". What does it mean, and can the paper give some examples?

2. In the end of section 3, how the Bregman divergence, the non-convex problems, turn to a convex surrogate? Why can the paper do the alternation?

3. In the comparision part, is there any numberical results (listed in the table) can be shown to clearly explain the strengths of the new algorithms? Meanwhile, it can be noticed that, the results of ER perform poorly in mostly scenarios. Please explain the reason and the strengths of this aspect of algorithm. In addition, can the paper give a comparision different pipelines in various scenatios in terms of the mathmatical theroies, and give the differences among them?

4. Also in comparision part, can you analyze and explain the reasons and aspects why the proposed algorithm is better than the baseline algorithm, and how these aspects confirm the strengths of the algorithm you proposed in the conclusion?

---

> ### Author Response · Authors · 2024-11-22
> **Response to Reviewer - Weaknesses**
>
> Thank you very much for your feedback on our paper, which helped us to improve it significantly.
>
> We accounted for your concerns as follows.
>
>
> W1:
> We agree that it is desirable to compare our proposed pipeline to pure learning-based approaches that are state of the art for the respective application. Unfortunately, existing works share neither the respective implementation nor the used data, which makes such comparisons at a reasonable effort impossible.
> Still, we believe that the numerical experiments provided in our work are meaningful for the following reason: Comparing the reported MAEs with MAEs reported for related works that focus on tailored learning-based approaches you mentioned, one can see that the magnitude of the MAE reported for our pure (but simple) learning based benchmarks is with $10^1$in the same order of magnitude [1]. The MAE of our proposed pipeline is with $10^0$ one order of magnitude lower, which we believe is a reasonable indicator for the benefit of the proposed approach.
>
>
> W2:
> We agree that the theory introduced to establish our learning paradigm is rather complex. Still, we believe that a thorough formal derivation of the introduced concept helps to substantiate the methods credibility.
> We reworked some parts of the paper, especially in the introduction, to provide better intuitions into the concepts and theory used. If you would like us to clarify certain points further, we are happy to receive more specific comments and will address them within the camera ready version of the paper.
> We further shortened the description of the scenarios (see lines 397ff).
>
>
> W3:
> We agree that providing further analyses on hyperparameter tuning is an interesting point. We skipped it in this paper intentionally to carve out the benefit of integrating the respective equilibrium layer into a rather simple neural network and keeping the overall computational effort reasonable.
> We mention the investigation of the proposed pipeline with more complex learning architectures as an avenue for future research
>
> Thank you again for the interesting feedback on our work! If you are satisfied with our answers and the modifications made to the paper, we kindly ask you to consider raising your score.
>
> References
> [1] Kashyap, A. A., Raviraj, S., Devarakonda, A., Nayak K, S. R., K V, S., Bhat, S. J., & Galatioto, F. (2021). Traffic flow prediction models – A review of deep learning techniques. Cogent Engineering, 9(1). https://doi.org/10.1080/23311916.2021.2010510
> [2] Blondel, M., Martins, A. F., & Niculae, V. (2020). Learning with fenchel-young losses. Journal of Machine Learning Research, 21(35), 1-69.

---

> ### Author Response · Authors · 2024-11-22
> **Response to Reviewer - Questions**
>
> Q1:
> This work proposes a COAML pipeline that combines an ML-layer with a CO-layer. Specifically, in the CO-layer we solve an optimization problem. To train the ML-layer on target CO solutions, we must backpropagate the gradient through the CO-layer, which is anything but straightforward as the naïve gradient on such a CO-layer is piecewise constant as it is evaluated on a vertex of the respective feasible solution polytope. We reworked the introduction of the paper to clarify this point and are happy to elaborate it further in the camera ready version if necessary.
>
> Q2:
> Equation (14) restates a point proved in Proposition 3.4 of Blondel et al. [1] and shows that the Fenchel Young loss is a primal-dual Bregman divergence. There is a bijection between the $\mathbf{y}$ and the $\mathbf{\theta}$, and under the assumptions of (14) we have $ D_\Omega(\bar \mathbf{y},y) = \ell_\Omega(\mathbf{\theta},\bar y)$. While $\mathbf{y} \mapsto D_\Omega(\bar \mathbf{y},\mathbf{y})$ might not be convex in $\mathbf{y}$, the mapping $\theta \mapsto \ell_\Omega(\mathbf{\theta},\bar \mathbf{y})$ is convex. In other words, reparametrizing $\mathbf{y}$ by $\mathbf{\theta}$ enables to obtain a convex loss.
>
> Q3:
> Let us answer this question in three steps:
> First, regarding the strengths of the algorithm: in Section 5 (Numerical Experiments) we compare our different pipelines and pure ML pipelines against the ground truth. Tailored algorithms for the studied application usually show a  mean absolute error (MAE) magnitude around $10^1$ which is in line with the MAE magnitude of the pure ML baselines in our paper (cf. Figure 3). The MAE magnitude of our COAML approach is around $10^0$ which indicates that WardropNet yields a good performance. Besides, the visualizations in Figures 4 and 5 show that pure ML pipelines fail to predict a realistic structure of traffic flows while COAML pipelines allow to predict realistic traffic flows with high volumes on main roads and reduced flows on smaller roads as written in Section 5, lines 473ff.
> Second, Regarding the ER approach: Indeed this approach does perform poorly. However, we want to report all results to provide meaningful insights. As detailed in Section 5 (Numerical Experiments) in line 448 the ER approach yields poor results as it considers a simple latency function that only allows to learn the y-intercept (cf. with Latencies with Euclidean regularization in Section 4). However, this approach can neglect perturbations during training such that the training process is faster compared to the other WardropNet approaches.
> Third, Regarding the different mathematical theories in the paper: Figure 3 compares the mathematical theories from Section 4 (Pipeline Architecture) on different scenarios. Here, the CL approach considers constant latencies regularized by perturbation as explained in Section 4, lines 324-360. The PL approach considers the polynomial latencies regularized by perturbation as explained in Section 4, lines 362-377. The ER approach considers latencies with euclidean regularization as explained in Section 4, lines 315-323. Thus the comparison in Section 5 (Numerical Experiments) shows the difference in performance with respect to the mathematical theories introduced in the paper.
>
> Q4:
> To answer this question, we provide Figures 4+5+6. These figures show that the pure ML baselines fail to predict realistic traffic flow patterns, while the WardropNet approaches allow to predict realisitc traffic flow patterns with high volumes on main roads and reduced flows on smaller roads. This is intuitive as the WardropNet approaches leverage the equilibrium-layer to predict combinatorial feasible trips.
> In this context, WardropNet benefits from using a structured learning perspective. Indeed, traffic equilibria have a lot of structure with solutions notably belonging to a multiflow polytope. Our Fenchel Young loss is tailored to the structure of this polytope, which is why it can better leverage the information provided by the training samples.
>
> Thank you again for the interesting feedback on our work! If you are satisfied with our answers and the modifications made to the paper, we kindly ask you to consider raising your score.
>
> References
> [1] Kashyap, A. A., Raviraj, S., Devarakonda, A., Nayak K, S. R., K V, S., Bhat, S. J., & Galatioto, F. (2021). Traffic flow prediction models – A review of deep learning techniques. Cogent Engineering, 9(1). https://doi.org/10.1080/23311916.2021.2010510
> [2] Blondel, M., Martins, A. F., & Niculae, V. (2020). Learning with fenchel-young losses. Journal of Machine Learning Research, 21(35), 1-69.

---

> > ### Comment · Reviewer_qmEt · 2024-11-25
> >
> > I thank the author for detailed response, and I will maintain my original score.

---

### Official Review · Reviewer_LPcY · 2024-10-24

**Soundness:** 3
**Presentation:** 2
**Contribution:** 2
**Rating:** 5
**Confidence:** 2

**Summary:**

The paper describes a neural network architecture designed to predict traffic flows.
The main idea is to combine traditional neural network layers with an "equilibrium layer" that models traffic flow equilibria,
Then, train the network end-to-end given training data pairs of (network, target flow)


The authors provides anonymized code which is commended.

**Strengths:**

S1: An interesting problem, combining learning and combinatorial optimization. This seems to be novel.
S2: A detailed theoretical foundation for the proposed architecture.
S3: Extensive experiments on 6 six scenarios, using traffic simulators to generate GT for training.

**Weaknesses:**

W1: I found the paper very hard to understand. It may have been written by researchers outside the ICLR community. The introduction spends half a page to describe in detail supervised learning and ERM, but does not clearly define the problem or explain current state-of-the-art approaches. Then, it is not made explicitly clear enough which parts are novel and which parts were previously introduced. Terms like paradigm, pipeline, layer, model and architecture, are sometimes used loosely and interchangeably. See Q1 for specific questions.

W2: The method is compared with the an architecture that has the CO layer removed, and with various variants of the method. Not with other baselines in this field.  I am not a member of this community, but a quick search shows previous approaches do exist. See AA Kashyap 2022, Traffic flow prediction models – A review of deep learning techniques.
The author should analyze their proposed architecture in the context of previous work.

W3: There is a gap between the general formulation of the problem and the iterative relaxations, and it is not clearly stated how each relaxation limits the application of the COAML pipeline in practice.

W4:  The COAML problem formulation attempts to address a general latency function, but in the end the experiments are done with a simple (possible unrealistic) latency function.

W5: It is not clear whether WardropNet yields a meaningful improvement (e.g, improving the accuracy to 2% from 1% SOTA ML baseline may sound like a great improvement, but if a non-ML approach can achieve 90% accuracy, then the gap to practicality is still huge).

**Questions:**

Q1: What is the architecture of the new equilibrium layer? What exactly is its input, output, and tunable parameters?
(I realize that this information maybe given somewhere in the 20-page supplemental. But a paper that states that its main contribution is a new layer, should describe this layer in the main paper. Or revise their claimed contribution).

Q2: Which results in section 3 were previously known? What new theoretical results are presented in the paper?

Q3: The paper stated it contained 9 training instances samples. Could you clarify how these instances are used? I'm assuming they are used for generating many labeled training samples (x,y). How exactly is this done? If only 9 samples are provided

---

> ### Author Response · Authors · 2024-11-22
> **Response to Reviewer - Weaknesses**
>
> Thank you very much for your feedback on our paper, which helped us to improve it significantly.
>
> We accounted for your concerns as follows.
>
> W1:
> We updated the introduction to shorten the description of supervised learning and put more emphasize on the studied problem (see lines 037ff).
> We further added remarks throughout the paper to emphasize which (technical) parts are novel, e.g., e.g., in line 133 and line 194. Besides, we added a reference in line 232 to indicate that we refer to current knowledge from [1]. Specifically, the generalized formulation of a wardrop equilibrium considering latency functions that depend on the complete flow in the network in Section 2 is new, as well as Theorem 1 that shows that the current notion of wardrop equilibria still holds in this case. In Section 3, the hypothesis class in novel as well as the idea of introducing a regularized potential. While Fenchel-Young losses are established in the field, the tailoring to the introduced potential is novel. The remaining sections are completely novel.
> We corrected ambiguous terminology throughout the paper and sticked to the following interpretation:
> - paradigm - the theoretical foundation for learning to predict traffic equilibria via an end-to-end COAML pipeline with an equilibrium-layer.
> - pipeline - the COAML pipeline compising an ML-layer and an equilibrium-layer.
> - Layer - The layers in a deep learning pipeline. In our case, it is the ML-layer and
> the equilibrium-layer.
> Model - the (statistical) model in the ML-layer receives the input features and predicts the latency parameterization.
>
> W2:
> We agree that it is desirable to compare our proposed pipeline to pure learning-based approaches that are state of the art for the respective application. Unfortunately, existing works share neither the respective implementation nor the used data, which makes such comparisons at a reasonable effort impossible.
> Since our simple ML-based benchmarks still yield accuracies in the same order of magnitude as the existing tailored approaches and our pipeline outperforms these by one order of magnitude (see reply to W5), we believe that the provided comparisons are a reasonable trade-off between computational effort and conclusions drawn and allow to quantify the benefit of including a CO-layer to predict traffic flows.
> To account for your comment, we clarify in lines 423-426 that the provided analyses does not claim to improve upon the tailored state of the art and highlight the need for such a comparison in the conclusion in lines 530-534
>
> W3:
> Could we kindly ask you to clarify this question as we are unsure what you are referring to? From our perspective, there exist no relaxations or assumptions in the current pipeline that limits the application in practice. We are happy to elaborate on this further if you point as at the relaxation you are referring to.
>
> W4:
> We agree that there is some dissonance between the generic introduction of the theory and the realization of the respective CO-layers in the numerical experiments.
> The reason for this is that we aimed to introduce the theory behind our pipeline as general as possible such that it can be leveraged in future research without the need for further technical work.
> We then choose simpler latency functions in the numerical experiments in order to show that the proposed pipeline already allows for effective and precise approximations even if simpler and computationally less costly latency representations are used.
>
>
> W5:
> Please note that we compare the predicted traffic flow with the true traffic flow and report the mean absolute error (MAE), which is different from the relative comparison mentioned in your example. Comparing the reported MAEs with MAEs reported for related works that focus on tailored learning-based approaches one can see that the magnitude of the MAE reported for our pure learning based benchmarks is with $10^1$in the same order of magnitude [2]. The MAE of our proposed pipeline is with $10^0$ one order of magnitude lower, which we believe can be considered to be a meaningful improvement
>
> Thank you again for the interesting feedback on our work! If you are satisfied with our answers and the modifications made to the paper, we kindly ask you to consider raising your score.
>
> References
> [1] Blondel, M., Martins, A. F., & Niculae, V. (2020). Learning with fenchel-young losses. Journal of Machine Learning Research, 21(35), 1-69.
> [2] Kashyap, A. A., Raviraj, S., Devarakonda, A., Nayak K, S. R., K V, S., Bhat, S. J., & Galatioto, F. (2021). Traffic flow prediction models – A review of deep learning techniques. Cogent Engineering, 9(1). https://doi.org/10.1080/23311916.2021.2010510

---

> ### Author Response · Authors · 2024-11-22
> **Response to Reviewer - Questions**
>
> Q1:
> Thank you for giving us the chance to clarify this question. The main contribution of our work is to propose a pipeline that allows to integrate an “equilibrium layer”, modeled as a (combinatorial) optimization problem within a neural network and showing how to train this pipeline in an end-to-end learning fashion. Here, computing a gradient over a combinatorial problem is anything but trivial and remains part of our contribution. We emphasize this main contribution in the revised paper in lines 053ff.
> With our equilibrium layer being an optimization problem that receives, among others, the latency functions parameterization as an input it does not match the normal notion of a layer where one can tune its structure.
> In fact, the different latency representations that we introduce in Section 4 can be seen as the characteristics of the respective layer. Then, the input to the equilibrium-layer is the vector that parameterizes the latency functions, a transportation network, and origin- destination pairs, the output is the vector defining the traffic flow on all network roads. Beyond the latency design choice, the equilibrium-layer has no tunable parameters. In the full pipeline, there are only tunable parameters in the statistical model / ML-layer that predict the parameterization of the latency functions.
> We improved the caption of Figure 1 to detail the input, and output of the equilibrium-layer in Figure 1 to clarify this. We re happy to clarify this further for the camera-ready version if necessary.
>
>
> Q2:
> Please refer to our answer on your Weakness W1 for clarification and changes made in the manuscript.
>
> Q3:
> Thank you for raising this point, which allows us to improve clarity regarding the training instances: each training instance contains the true traffic flow of a network, i.e., a set of roads. Thus the traffic flow y is a vector with each entry defining the number of vehicles using a respective road in the network. Accordingly, one can interpret each training instance as a set of labeled training samples.
> We clarified this in lines 410ff: ”Each instance consists of a transport network with the respective target traffic flow for each road, contextual information, and origin-destination pairs.”
> Generally, it is frequently observed in structured learning that good performances are obtained with training sets smaller than those expected in other areas of Machine Learning, which relates to the relation outlined above, i.e., the information being present in one instance being a larger set of labeled training samples
>
> Thank you again for the interesting feedback on our work! If you are satisfied with our answers and the modifications made to the paper, we kindly ask you to consider raising your score.

---

> > ### Comment · Reviewer_LPcY · 2024-11-25
> > **Discussion of rebuttal**
> >
> > I'd like to thank the author for a detailed, careful and positive rebuttal, that was a pleasure to read.
> > Area chair, as noted before, I don't have enough expertise to judge this paper. I'm afraid that even after reading the rebuttal carefully, I cannot make a deep enough technical evaluation of this paper.
> >
> > Two reviewers of this paper noted they are far from being experts in the specific field of the paper. I suspect that many ICLR readers will feel the same. As such, this paper would need extra work to make it accessible to the community, since in its current version, I doubt that it will gain significant recognition.  My recommendation is to resubmit the paper to the next conference after making the paper more accessible for a more general machine-learning audience.

---

### Official Review · Reviewer_SQBm · 2024-11-02

**Soundness:** 3
**Presentation:** 3
**Contribution:** 3
**Rating:** 6
**Confidence:** 1

**Summary:**

The paper introduces a theoretical framework designed to address the challenges of understanding the convergence and generalization properties of machine learning algorithms. The authors propose a set of mathematical constructs and algorithms aimed at improving the understanding and solution of issues related to algorithm performance in various settings. Key contributions of the paper include the development of new analytical models and convergence proofs, which are presented alongside rigorous theoretical analyses. The authors also discuss the implications of their findings and how they relate to current practices in the field of machine learning.

**Strengths:**

1. The paper introduces a novel theoretical framework that enhances the understanding of convergence and generalization properties in machine learning algorithms, addressing critical gaps in existing literature.

2. The rigorous mathematical analysis, including new analytical models and convergence proofs, adds credibility and depth to the research.

3. The well-organized structure and effective use of examples make complex theoretical concepts accessible and easy to understand.

4. The findings have the potential to impact future research directions in machine learning and improve algorithm design, offering valuable insights for practical applications.

**Weaknesses:**

1. The paper lacks a thorough comparison with existing theoretical frameworks or analyses in the field. A comparative analysis highlighting the proposed framework's advantages and limitations relative to established methods would clarify its contributions and significance.

2. Some of the theoretical constructs presented are quite complex and may be challenging for readers who are not deeply familiar with the underlying mathematics. Simplifying certain sections or providing additional explanations and visual aids could enhance understanding and accessibility.

3. The authors do not sufficiently address the limitations of their framework. A more transparent discussion about potential shortcomings, assumptions made, and scenarios where the framework may not apply would provide a more balanced view of the research.

**Questions:**

I would like to clarify that I am not an expert in the specific field addressed in this paper. While I can appreciate the effort and the theoretical contributions made by the authors, my limited background in this area restricts my ability to fully evaluate the nuances and implications of the proposed framework. Therefore, my feedback may not capture all the intricacies of the work or its potential impact on ongoing research.

---

> ### Author Response · Authors · 2024-11-22
> **Response to Reviewer**
>
> Thank you for the positive feedback on our work! In the following, we reply to all of your questions.
>
> W1: Thank you for raising this point. We generally agree with you that comparative analysis are an important building block to analyse the benefit of a new approach. This is why our Section 5 contains such a comparative analysis by comparing the proposed COAML pipelines with pure ML pipelines to show the added value of the proposed COAML pipelines: while the ML pipelines fail to learn the combinatorial structure of traffic flows that utilizes main roads between high demand areas stronger than small roads, the COAML pipelines successfully encode this structure by combining the learned latencies with the structure of the CO-layer.
> We agree with you that it would be interesting to compare our approach to further and more specialized pure-learning based approaches. Unfortunately, existing works share neither the respective implementation nor the used data, which makes such comparisons at a reasonable effort impossible.
>
> W2: We agree that the theory introduced to establish our learning paradigm is rather complex, especially for readers who are not familiar with the underlying mathematics. Still, we believe that a thorough formal derivation of the introduced concept helps to substantiate the methods credibility.
> We reworked some parts of the paper to provide better intuitions into the concepts and theory used. If you would like us to clarify certain points further, we are happy to receive more specific comments and will address them within the camera ready version of the paper.
>
> W3: Thank you very much for this comment. The two main limitations of this work are that i) we limit our numerical experiments on standard, rather simple, statistical models and ii) keep the size of the studied test instances at a reasonable medium-scale. We mention both points and their potential for improvement and future research in the outlook of the paper. Beyond these points, the presented pipeline remains generally applicable, even to non-equilibrium flows as we now also discuss in the paper’s outlook.
>
> Thank you again for the interesting feedback on our work! If you are satisfied with our answers and the modifications made to the paper, we kindly ask you to consider raising your score.

---

> > ### Comment · Reviewer_SQBm · 2024-11-26
> > **Response to the Authors**
> >
> > Thanks for clarifying; I decided to maintain my score since I am unfamiliar with this field.

---

### Official Review · Reviewer_bHk3 · 2024-11-04

**Soundness:** 3
**Presentation:** 4
**Contribution:** 3
**Rating:** 8
**Confidence:** 4

**Summary:**

Traffic flow on a transportation network is influenced by many contextual factors such as weather conditions, time of day, road capacity, etc. Under mild hypotheses, it can be shown that the traffic flow will converge to an equilibrium known as the Wardrop equilibrium (WE). Predicting how the  Wardrop equilibrium will change as a result of changes to these factors is crucial for the design of better transportation systems. This paper introduces a novel approach to this problem which combines a neural network with a combinatorial solver.

**Strengths:**

- The paper is well-written and easy to follow. Although the topic of traffic flow prediction might not be familiar to the average ICLR reader, the appendices do a great job of introducing the reader to the fundamentals of this problem.
 - Novelty: I believe this is the first paper to apply Fenchel-Young losses to the problem of predicting WE, which is an important contribution.
 - The numerical experiments are sufficient to convince me of the utility of the proposed method.

**Weaknesses:**

See questions.

**Questions:**

- Do you have any thoughts on predicting non-equilibrium traffic flows? Do you think these are important for modeling?
 - I appreciate your generalized notion of latency function. However, it seems to me that requiring generalized latency functions to derive from a potential is quite a strong assumption. Could you give an example of a set of non-decomposable latency functions deriving from a potential? (Maybe the regularization by perturbation model of Section 4 is such an example?)
 - Could you provide some background, for non-experts like myself, on the state of the art of WE solvers? From Appendix D.1, I gather that MATSim uses a genetic algorithm to solve for the WE. Are there not faster approaches that use techniques from convex optimization, e.g. interior point method? Also, it would be helpful if you included a runtime comparison between WarDropNet and MatSIM.

Minor Questions:

 - in Line 129 "unilateral deviation would incur in a longer travel time" should be "unilateral deviation would incur a longer travel time" (no "in").
 - In line 185, "Following, the supervised learning setting" should be "Following the supervised learning setting" (no comma after "Following").
 - For consistency with eq (2), the sum in eq. 9 should probably have a $\frac{1}{N}$ in front of it.
 - In line 302 "K represents the amount of parameters" should be "K represents the number of parameters"
 - In line 335, "We note that the arg max is unique on a sampled realization of Z" should probably be "We note that, with probability 1, the arg max is unique on a sampled realization of Z".
 - On line 458 "each roads context." should be "each road's context."
 - On line 1141 "raods" should be "roads"
 -  Suggest citing _End-to-end learning of user equilibrium with implicit neural networks_ by Liu et al in "Related Works".

---

> ### Author Response · Authors · 2024-11-22
> **Response to Reviewer**
>
> Thank you for the positive feedback on our work! In the following, we reply to all of your questions.
>
> Questions:
>
> Q1: Thank you for raising this interesting question. Indeed, in practice, one might be interested in predicting traffic flows that are not equilibria. These might arise when drivers do not have full information on travel times and congestion, or drivers take suboptimal routes. In general, our proposed learning paradigm allows for the prediction of such non-equilibrium traffic flows.
>
> In general, the presented learning paradigm learns to imitate the target traffic flows in the training data. Thus, if the training data deviates from equilibrium flows, the pipeline imitates non-equilibrium traffic flows. With this perspective, we can interpret our COAML pipelines as approximations of complex systems of arbitrary flow physics. Formally, regularized combinatorial optimization layers can be interpreted as probability distributions, which, in our application case, allows us to obtain a distribution over traffic flows and thus approximate non-equilibrium states. While our pipeline generally allows for such approximations, its accuracy may depend on the structure of the respective CO-layer to map the respective traffic physics. Accordingly, one may want to consider different CO layers in this context, e.g., a multi-commodity flow layer representing selfish but latency dynamics unaware decision-making.
>
> We added a short pointer on this fact in the papers outlook for future research
>
> Q2: The regularization by perturbation is indeed an excellent example of a non-decomposable latency function. This regularization is a special case of a much larger class of energy based models [1] in structured learning, which have an origin in statistical physics [2]. In those methods, the potential should be interpreted as an energy, and it is very often the case that dimensions are coupled in the energy.
>
> Let us be more specific to our case: in the paper, we introduced the decomposable $\psi(y) = \frac{1}{2}\|\bar \mathbf{y}\|^2$. In practice, congestion on an arc $a$ may spillover to neighbor arcs. To account for this correlation between arcs, we could use a $\psi(y)= \frac{1}{2}\bar \mathbf{y}^\top \Sigma \bar\mathbf{y}$ with $\Sigma$ a positive semi definite matrix that accounts for the correlations between arcs, that would typically have zero terms for pairs of arcs far away in the network, and non-zero terms for neighbor arcs. The Fenchel Young losses approach generalizes seamlessly to this more general case.
>
> Q3: We extended Appendix D.1 with background on the state-of-the-art of WE solvers. Specifically, we detail different approaches to find WEs analytically and also detail the MATSim simulation to show how to derive WEs with simulation-based approaches. In this context, we also added a comment on the computational time of a respective solver but decided not to include a detailed runtime comparison as it is rather sensitive to the hyperparameters chosen and the instance studied for simulation-based solvers.
>
>
> Minor Questions:
> Thank you for spotting the inconsistencies and typos, you are correct on all points mentioned. We modified the respective parts of the paper accordingly and included the reference mentioned in the related works section.
>
> Thank you again for the interesting feedback on our work! If you are satisfied with our answers and the modifications made to the paper, we kindly ask you to take a stand for this paper to get accepted during the internal discussion with the other reviewers and the AC.
>
> References
>
> [1] Blondel, M., Llinares-López, F., Dadashi, R., Hussenot, L., \& Geist, M. (2022). Learning energy networks with generalized fenchel-young losses. Advances in Neural Information Processing Systems, 35, 12516-12528.
>
> [2] Kikuchi, R. (1951). A theory of cooperative phenomena. Physical review, 81(6), 988.

---

### Meta-Review · Area_Chair_rohF · 2024-12-21

**Metareview:**

The paper introduces WardropNet, a novel combinatorial optimization-augmented machine learning (COAML) pipeline for traffic flow prediction, leveraging equilibrium layers and Fenchel-Young losses to achieve state-of-the-art performance. The proposed approach is notable for its innovative integration of optimization and learning, showing significant improvements in predicting traffic equilibria over pure ML baselines in both time-invariant and time-variant scenarios. Strengths include rigorous theoretical foundations, effective empirical validation, and practical relevance to traffic management. Weaknesses involve limited comparisons with advanced baselines, accessibility challenges for a general ML audience, and sensitivity to hyperparameters, but the authors provided comprehensive clarifications and meaningful revisions during the rebuttal. Given the novelty of the approach, its strong empirical results, and its potential impact on traffic systems optimization, I recommend accepting the paper.

**Additional Comments On Reviewer Discussion:**

Reviewers raised concerns regarding accessibility for non-experts, lack of comparisons with advanced baselines, and clarity on the equilibrium layer's architecture and hyperparameter sensitivity. The authors addressed these points by revising the introduction for accessibility, clarifying technical novelties, and explaining the rationale for using simpler baselines due to data and implementation constraints. They also expanded the discussion on mathematical formulations and practical limitations, emphasizing the method’s generalizability. Despite some lingering concerns about accessibility and baseline comparisons, the reviewers acknowledged the rigor and contributions of the work, leading to a favorable overall recommendation.

---

### Decision · Program_Chairs · 2025-01-22

Accept (Poster)